# Inhibitors of Brassinosteroid Biosynthesis and Signal Transduction

**DOI:** 10.3390/molecules24234372

**Published:** 2019-11-29

**Authors:** Wilfried Rozhon, Sonia Akter, Atiara Fernandez, Brigitte Poppenberger

**Affiliations:** Biotechnology of Horticultural Crops, TUM School of Life Sciences Weihenstephan, Technical University of Munich, Liesel-Beckmann-Straße 1, 85354 Freising, Germany

**Keywords:** bikinin, brassinosteroid biosynthesis, brassinosteroid signaling, brassinazole, gibberellins, inhibitors, pyrimidines, sterols, triazoles

## Abstract

Chemical inhibitors are invaluable tools for investigating protein function in reverse genetic approaches. Their application bears many advantages over mutant generation and characterization. Inhibitors can overcome functional redundancy, their application is not limited to species for which tools of molecular genetics are available and they can be applied to specific tissues or developmental stages, making them highly convenient for addressing biological questions. The use of inhibitors has helped to elucidate hormone biosynthesis and signaling pathways and here we review compounds that were developed for the plant hormones brassinosteroids (BRs). BRs are steroids that have strong growth-promoting capacities, are crucial for all stages of plant development and participate in adaptive growth processes and stress response reactions. In the last two decades, impressive progress has been made in BR inhibitor development and application, which has been instrumental for studying BR modes of activity and identifying and characterizing key players. Both, inhibitors that target biosynthesis, such as brassinazole, and inhibitors that target signaling, such as bikinin, exist and in a comprehensive overview we summarize knowledge and methodology that enabled their design and key findings of their use. In addition, the potential of BR inhibitors for commercial application in plant production is discussed.

## 1. Introduction

Brassinosteroids (BRs) are a group of polyhydroxylated steroidal plant hormones involved in many developmental processes including hypocotyl elongation [1,2], root growth [3,4,5,6], stomata patterning [7,8,9], pollen development and germination and pollen tube growth [10], tracheary element differentiation [11,12,13], xylem formation [13,14,15,16,17], and senescence [2,18]. In addition, they are involved in adaptation to environmental cues including drought [19,20], cold [21,22,23], heat [24,25,26], salinity [27,28] and shortage of nutrients, particularly iron [29]. BR deficient plants are dwarfed and exhibit dark green, downward curled leaves, shortened petioles, hypocotyls and internodes and show delayed flowering and reduced male fertility. In contrast, plants overaccumulating BRs show increased height and elongation of hypocotyls and petioles. The same phenotypes can also be obtained by external application of BRs. Among the BRs only brassinolide (BL), an end product of BR biosynthesis, and its precursor castasterone (CS) are bioactive [30]. While CS seems to be ubiquitously present in the plant kingdom, BL has not been detected in non-flowering plants including ferns and mosses [31], and in many monocots including rice [32,33].

### 1.1. BR Biosynthesis

Like all steroids, BRs are derived from the isoprenoid squalene (Figure 1). Squalene is cyclized by action of squalene epoxidases (SQEs; Table 1) and cycloartenol synthase (CAS1) to cycloartenol, the first committed precursor in plant sterol biosynthesis. Cycloartenol is further metabolized to 24-methylenelophenol, where phytosterol biosynthesis divides in two branches. The C29 branch has β-sitosterol and stigmasterol as end products. The C28 branch leads to campesterol, but in algae and some land plants like *Arabidopsis thaliana* also to brassicasterol. Stigmasterol, β-sitosterol and campesterol are bulk phytosterols and regulate membrane fluidity and permeability of the cell membrane [34]. In addition, campesterol also serves as a precursor for BR biosynthesis. Interestingly, the *A. thaliana* mutants *dwf1* [35,36], *dwf5* [37], *dwf7* [38] and *smt2* [39], which are all defective in sterol biosynthesis downstream of 24-methylenelophenol, show phenotypes like BR biosynthesis mutants and can be rescued by application of bioactive BRs. This indicates that in these mutants, hormone deficiency has a higher impact on the phenotype than the altered sterol profile [40]. In contrast, mutants with defects upstream of 24-methylenelophenol, including *smt1* [41], *cyp51a2* [42], *fk* [43,44] and *hyd1* [45], exhibit severe defects in embryogenesis, which cannot be rescued by application of BL. This may indicate that sterols may have BR-independent functions in embryogenesis [40].

BR biosynthesis was elucidated by a combination of genetic approaches [15,18,53,55], rescue experiments of mutants with putative precursors [51,52], analysis of intermediates [51,56,57] and enzymatic activity assays [49,58,59,60,61]. Screening and analysis of mutants allowed identification of most enzymes involved in BR biosynthesis [62]. In many studies *A. thaliana* was used but the analysis of tomato (*Solanum lycopersicum*), pea (*Pisum sativum*) and rice (*Oryza sativa*) mutants also provided important results [62]. Based on these results, BR biosynthesis was initially proposed to act in two parallel pathways, the late and the early C-6 oxidation pathway, highlighted in green and in blue in Figure 1, respectively. However, detailed biochemical studies provided convincing evidence that these pathways are not the main routes of BR biosynthesis. Enzymatic studies with heterologously expressed DWF4 showed high activity for hydroxylation of campesterol to (*22S*)-22-hydroxycampesterol while its catalytic efficiency for hydroxylation of campestanol to 6-deoxocathasteone was more than 300 times lower [61]. Importantly, DWF4 could not hydroxylate 6-oxocampestanol to form cathasterone, the first presumed step of the late C-6 oxidation pathway. Enzymatic studies with CPD showed that it cannot utilize campesterol. In contrast, CPD showed high activity for conversion of (*22S*)-22-hydroxycampesterol to (*22S*,*24R*)-22-hydroxyergost-4-ene-3-one. In addition, CPD could also catalyze several downstream reactions (Figure 1) [59]. ROT3 and its close homologue CYP90D1 showed high activity for 23-hydroxylation of 3-*epi*-6-deoxocathasterone, (*22S*,*24R*)-22-hydroxy-5a-ergostan-3-one and (*22S*,*24R*)-22-hydroxyergost-4-en-3-one, while they were almost inactive for hydroxylation of 6-deoxocathasterone [49]. Taken together, these data indicate that the early C-22 oxidation pathway, which is independent of campestanol, seems to be the predominant route in BR biosynthesis in *A. thaliana* (highlighted in pink in Figure 1). This pathway is interconnected with the late C-6 oxidation pathway at multiple steps. In contrast, the upper part of the early C-6 oxidation pathway from 6-oxocampesterol to teasterone seems to be of minor importance for synthesis of bioactive BRs in *A. thaliana* since these metabolites are not detected in this species [63]. This is also confirmed by the catalytic activities of *A. thaliana* BR6OX1 and BR6OX2, two cytochrome P450 monooxygenases with C-6 oxidation activity. Investigation of specificity revealed that these enzymes accept 6-deoxoteasterone, 6-deoxo-3-dehydroteasterone, 6-deoxotyphasterol and 6-deoxocastasterone as substrates while campesterol is not oxidized [60]. BR6OX2 but not BR6OX1 has also a second activity: it catalyzes the Bayer-Villiger-type oxidation of CS to BL [30,60,64], the end product of BR biosynthesis in most species. Interestingly, monocots have only one copy of a BR6OX gene while dicots have two or more copies, providing additional evidence that CS might be the final product of BR biosynthesis in many monocots including rice [32]. However, it must be mentioned that BL has been detected in a few monocots, for instance in *Lilium elegans* [65].

While most enzymes of BR biosynthesis have been identified, at least two remain to be determined. The first is involved in epimerization of C-3, where the 3-hydroxy group is oxidized to a keto group, which is subsequently reduced back to a 3-hydroxy group but with opposite stereochemistry. This reaction sequence is illustrated, for instance, by the reaction of 6-deoxoteastereone to 6-deoxo-3-dehydroteasterone and finally to 6-deoxotypahsterol (Figure 1). The first step is catalyzed by CPD. However, the second step must be catalyzed by another enzyme since CPD can oxidize 6-deoxo-3-dehydroteasterone back to its precursor 6-deoxoteastereone but not to 6-deoxotyphasterol, the product of this sequence [59] (Figure 1). This provides compelling evidence that a second, yet unknown enzyme must be involved in C-3 epimerization.

The second step is C-2 hydroxylation of 6-deoxotyphasterol and typhasterol to 6-deoxocastasterone and CS, respectively. In pea this reaction is catalyzed by PsDDWF1 (CYP92A6), a cytochrome P450 monooxygenase, and a supplementary factor, Pra2 [66]. However, their homologues in *A. thaliana* remain elusive.

Recently, it was shown that overexpression of DWF1 in the *det2* background rescued the BR-deficient phenotype to some extent. In addition, in vitro assays revealed that 6-deoxydolichosterone and dolichosterone can be converted by DWF1 to 6-deoxocastasterone and castasterone [67]. These data suggest that alternative pathways for synthesis of bioactive BRs may exist. However, further studies are required to estimate the contribution of these putative pathways for production of bioactive BRs.

Concentrations of substrates and products of individual steps of the C-22 oxidation pathway are present in *A. thaliana* in a ratio close to 1:1. Exceptions are hydroxylation of campesterol to 22-hydroxycampasterol, the first step of BR biosynthesis catalyzed by DWF4, and conversion of 6-deoxocastasterone to CS catalyzed by BR6OX1 and BR6OX2, where the ratios are approximately 50000:1 and 7:1, respectively [63]. This indicates that DWF4 might catalyze the main rate limiting step in BR biosynthesis. This is substantiated by the observation that plants overexpressing DWF4 show typical signs of BR overaccumulation including long hypocotyls, increased height of mature plants and yield of seeds [56]. In addition, the transcript levels of many BR biosynthesis genes are feedback-controlled by BR signaling [68,69] with BR6OX2, CPD, CYP90D1, DWF4 and ROT3 being most severely affected [68,70,71].

So far, many compounds interfering with BR biosynthesis have been identified, most of which target either DWF4 or CYP90D1 [72,73,74,75]. Inhibitors targeting the latter enzyme are also thought to inhibit ROT3, which acts redundantly with CYP90D1 [49], although direct evidence for that conclusion is still missing. In addition, a few compounds impeding 5α-reduction by inhibiting DET2 have been described [76,77]. Also voriconazole, a compound inhibiting CYP51A2 acting in sterol biosynthesis, causes BR deficiency [78].

### 1.2. Inactivation of BRs

Besides regulation of transcript levels, BR homeostasis is maintained by inactivation of BL and its precursors by modifying enzymes (Figure 2).

The two cytochrome P450 monooxygenases BAS1 [79] and SOB7 [80,81,82] mediate BR inactivation by hydroxylation of BL and its precursors. Overexpression of BAS1 and SOB7 results in typical signs of BR deficiency including dwarfism, shortened hypocotyls and dark green, downward curled leaves [79,80]. Binding assays with purified proteins showed that BAS1 preferably binds BL and CS [83], which is in agreement with results that it mainly hydroxylates these bioactive BRs on C-26 [84]. Also, the BAS1 homologue in tomato, CYP734A7, prefers BL and CS while its homologue in rice, CYP734A2, accepts a wide range of C-22 hydroxylated BRs as substrates [85]. Binding assays with SOB7 showed that it has high affinities for binding of teasterone and typhasterol, suggesting that it may preferably hydroxylate BL precursors [83]. However, the specific reactions catalyzed by SOB7 remain unknown [62].

Another means of BL and CS inactivation is 23-*O* glucosylation, which is catalyzed by the UDP-glucosyltransferases UGT73C5 and UGT73C6 [86,87]. Overexpression of both enzymes reduced levels of endogenous BRs and induced BR deficient phenotypes that could be partially rescued by application of exogenous BL [86]. Feeding experiments with overexpression and knock-down lines confirmed that UGT73C5 and UGT73C6 convert BL and CS to their glucosylated forms. Interestingly, there is first evidence that the glucosides were rapidly malonylated by a yet unidentified enzyme [87]. The functional significance of this modification remains to be shown.

A number of BRs can be modified by sulfation. Originally, a cDNA of a putative sulfotransferase was isolated from an *A. thaliana* cDNA library [88]. This clone was used to screen a genomic rape seed (*Brassica napus*) library, which allowed identification of three putative intronless sulfotransferases, designated BNST1 to 3, that were induced by salicylic acid at the transcriptional level. Heterologous expression in *E. coli* and in vitro enzyme assays revealed that BNST3 is capable of sulfating a number of sterols including 24-epiBL, 28-homoBL, 24-epiteasterone, 24-epicathasterone and the mammalian sex hormone β-estradiol. However, BL was not utilized [89] and overexpression of BNST3 did not result in any visible BR deficiency [90]. A fourth sulfotransferase isolated from *B. napus*, BNSt4, was also stereospecific for 24-epimers of BRs [90]. Similarly, AtSt1 and AtSt4a, two sulfotransferases isolated from *A. thaliana*, showed preference for 28-homobrassinolide [91] while BR and CS were sulfated only at a very low rate. In summary, although there is compelling evidence for a number of plant enzymes that can sulfate sterols in vitro, the biological function of these sulfotransferases remains elusive since overexpression in plants failed to produce any BR deficient phenotype. In addition, the in planta substrates of these sulfotransferases have not been identified so far.

Mutants with strong BR deficient phenotypes caused by overexpression of the acyltransferase DRL (also referred to BAT1 and PIZZA) were reported independently at the same time by three research groups [92,93,94]. The mutants were either identified by screening of FOX (full-length cDNA overexpressor) lines [92,93] or in an activation-tagged mutant population [94]. Transcription analysis revealed up-regulation of BR biosynthetic genes including CPD, DWF4, BR6OX2 and ROT3 and down-regulation of the BR inactivating genes BAS1 and SOB7. BAT1 expression was up-regulated by auxin [92] but repressed by ABA [94]. The effect of treatment with 24-epiBL was not consistent since one study reported increased BAT1 expression [94] while another study found no significant difference [92]. However, in the latter study a 100-fold lower concentration of 24-epiBL was used, which might explain the different observations. Profiling of BR intermediates showed that the mutants contained significantly less 6-deoxotyphasterol, typhasterol and 6-deoxocastasterone than the wild type. Only one study investigated the enzymatic activity in more detail and found that BAT1 can laurylate BL, CS and typhasterol [93]. Although the laurylated hydroxy group was not identified, it might be speculated that the 3-hydroxy group is modified by BAT1, since 3-*O*-lauryl, 3-*O*-myristyl and 3-*O*-palmityl BRs have been identified in BR-treated plant cell cultures [95,96].

Also the acyltransferase BIA1/ABS1 was independently shown by two research groups to inactivate BRs [97,98]. In both studies, mutants were isolated from *A. thaliana* activation tagging populations, since they showed extreme BR deficient phenotypes. Analysis revealed insertion of the activation tagging T-DNA upstream of gene *At4g15400*, which was confirmed to be highly expressed in the mutants [97,98]. BR profiling of the *bia1-1D* mutant showed that all BRs downstream of 6-deoxoteasterone were reduced, indicating that these intermediates might be modified by BIA1. Expression of *BIA1* was strongly enhanced by application of BL, which is common for metabolic enzymes. However, the enzymatic activity of BIA1 and the metabolites formed remain elusive.

Another BR catabolic enzyme with unknown activity is BEN1, a dihydroflavonol 4-reductase-like protein [99]. A possible reaction might be reduction of CS and typhasterol to their 6-hydroxy counterparts. However, these metabolites were not detected after feeding of *A. thaliana* seedlings with CS, indicating that the hydroxylated metabolite is either rapidly degraded or that BEN1 inactivates BRs by other means [62].

From the inactivation modes discussed above, 26-C hydroxylation is likely irreversible while glucosylation, sulfatation and acylation are potentially reversible, as it has been described for many plant hormones including auxin [100,101], salicylic acid [102,103,104] and abscisic acid [105,106]. Although the idea that such BR derivatives might act as storage forms is attractive, so far first evidence for reactivation of BRs from metabolites has only been provided for BR ester conjugates [96].

Currently, no inhibitors of BR inactivation have been reported. Given the huge number of BR inactivation modes, it is unlikely that a compound inhibiting a specific BR metabolic enzyme may induce any obvious morphological phenotype. In support, BR catabolic genes induce severe BR deficiency when overexpressed, while knock out mutants show no or very weak phenotypes, substantiating evidence for significant redundancy in BR inactivation.

### 1.3. BR Perception and Signal Transduction

In the last two decades, tremendous progress has been made in understanding how BRs are perceived and their signals are transduced to promote cellular responses (Figure 3). Bioactive BRs, particularly BL, bind with the C and D ring (Figure 2) and the aliphatic side chain to the cell-membrane localized receptor kinase BRI1 [107]. The A ring and parts of the B ring interact with the co-receptor BAK1 [108,109,110,111]. This triggers multiple auto- and transphosphorylation events between BRI1 and BAK1 [108,112,113,114], which causes activation of BRI1 and dissociation of BKI1 [115] from the cytoplasmatic domain of BRI1. In its free form, BKI1 antagonizes 14-3-3 proteins thereby enhancing accumulation of BES1/BZR1-type transcription factors in the nucleus [116]. In addition to its plasma membrane localization, signaling-competent BRI1/BAK1 hetero-oligomers were detected in the endocytotic compartment of plant cells, which might be important for proper BR signal transduction [117,118,119]. In special tissues the BRI1 homologues BRL1 and BRL3 might be responsible for BR perception [120,121]. After activation of BRI1, the signal is further transmitted by BSKs and CDG1 [122,123], protein kinases anchored by a palmityl residue to the cell membrane, to activate protein phosphatase BSU1 [124] and its homologues, which form homo- and heterodimers [125]. This, in turn, leads to inactivation of a number of GSK-3-like kinases. The best characterized member of these GSK-3-like kinases is BIN2, which is repressed by BSU1 by dephosphorylation of Tyr200 [124]. BIN2 was originally isolated independently by three research groups in mutant screens for plants with impaired BR signaling [126,127,128,129]. Analysis of the isolated alleles revealed that they are semidominant gain-of-function variants causing typical signs of BR deficiency including dark green downward curled leaves, shortened hypocotyls, partial de-etiolation, reduced plant height and low fertility. These findings identify BIN2 as a negative regulator of BR signaling. BIN2 knock out plants have a wild type-like phenotype since *A. thaliana* possesses 10 at least partially redundantly acting GSK-3-like kinases in total [123,124,130]. BIN2 and the other GSK-3-like kinases regulate the activity of BES1/BZR-like transcription factors by phosphorylation [2,69,131,132]. They are also important regulators of stress responses [133,134,135,136,137], although it remains elusive whether they act BR-dependent in those signaling events. The members of the BES1/BRZ1 family of transcription factors possess eight adjacent repeats of the sequence S/TXXXS/T, which is the consensus GSK-3β phosphorylation motif known from the animal field [131,138]. Phosphorylation of this domain impairs binding of BES1/BZR-type transcription factors to DNA [132]. Furthermore, phosphorylated BES1/BZR1-type transcription factors bind to 14-3-3 proteins, which promote their re-localization to the cytoplasm [139,140]. As mentioned above, the sequestrating effect of 14-3-3 proteins is probably antagonized by BKI1 that had been dissociated from the cytoplasmatic domain of BRI1. The dephosphorylated forms of BES1/BZR1-type transcription factors can bind either on their own [141,142] or together with basic helix-loop-helix transcription factors, for instance BIM1 [131], to promoters and regulate transcriptional activity of a huge number of genes [143,144]. PP2As, cytoplasmic serine-threonine protein phosphatases, counteract the activity of BIN2 by dephosphorylating BES1/BZR1-type transcription factors [145].

Besides regulating BES1/BZR1-like transcription factors, BIN2 and its homologues can also phosphorylate a number of other proteins including MYBL2, which, along with BES1 and BZR1, is important for down-regulation of BR biosynthesis [146]. Further targets include CESTA [147,148] and PIF4 [149,150], which, like HBI1 [151], induce BR biosynthesis and may thereby contribute to BR homeostasis.

A number of chemicals interfering with BR signal transduction have been identified. These compounds target either BRI1 or BIN2 and its homologues. For BRI1 several inhibitors have been identified by pharmacological approaches. In addition, one non-steroidal activator of BRI1 has been described. However, this molecule has quite a low potency compared to BL or synthetic steroidal BL analogs. Lithium is a well-known inhibitor of GSK-3s from animals and it has been shown to inhibit BIN2 and its homologues as well. Nevertheless, lithium is toxic for many plant species. With bikinin and its derivatives, BIN2 inhibitors with fewer side effects are now available.

## 2. Methods for Characterization of Small Molecules Interfering with BR Biosynthesis or Signaling

### 2.1. Determination of Potency

The potency of BR biosynthesis inhibitors is often measured by hypocotyl elongation assays using either cress (*Lepidium sativum*) or *A. thaliana* but also other plant species have been applied. While it has been established that hypocotyl growth reacts sensitively to BRs, it is important to note that also other plant hormones, for instance auxin [152,153,154] and gibberellin [155], and environmental conditions, for instance light intensity [156] and temperature [153], have major impacts on hypocotyl length.

For performing hypocotyl elongation assays plants are grown on solid media, for instance ½ Murashige and Skoog agar [157] containing 1% sucrose, supplemented with the compound to be tested. In case of cress, plates are usually incubated under long day conditions for 5–7 d prior measuring the hypocotyl length. A light intensity of 80 µmol∙s^−1^∙m^−2^ is suitable. Since the hypocotyl length reacts sensitively to light, care must be taken that all samples receive exactly the same light intensity. In addition, the temperature must be kept constant, usually at 21 °C. Hypocotyl assays using *A. thaliana* may also be incubated under long day conditions. While a temperature of 21 °C is also suitable for *A. thaliana*, the light intensity should be reduced to 20 µmol∙s^−1^∙m^−2^ for assaying BR biosynthesis inhibitors in order to obtain sufficiently elongated hypocotyls for the control. At higher light intensities, *A. thaliana* hypocotyls of the control are already very short and a further decrease of the length is hard to detect. In contrast, for compounds increasing hypocotyl length, for instance inhibitors of BIN2, *A. thaliana* is ideally grown at 80 µmol∙s^−1^∙m^−2^ for one week as these conditions allow optimal detection of increased hypocotyl elongation. Since *A. thaliana* hypocotyls typically have a length of a few mm, they are best measured using a microscope. Plates with *A. thaliana* can also be incubated in complete darkness. This has the advantage that ensuring identical light intensities can be omitted. However, prior to incubation a light pulse of at least a few hours should be applied to enhance germination. Hypocotyls of dark grown *A. thaliana* are well elongated and can thus be measured, for instance using a ruler. However, they are very fragile and must be handled with care. In addition, dark grown *A. thaliana* hypocotyls are shortened by BR biosynthesis inhibitors as well as by BL [158,159,160], which makes interpretation of the results sometimes more difficult.

According to our experience, hypocotyl assays with cress are simple and can be evaluated more rapidly. However, *A. thaliana* is often preferred since it is a widely used model plant. The reproducibility of both systems is comparable and the standard deviation (SD) is in the range of 10–20%. For obtaining reliable results at least 20 hypocotyls should be measured for each inhibitor concentration to be tested.

For a rapid comparison of the potency, hypocotyl elongation at just one concentration may be assayed. For more detailed characterization, the half maximal effective concentration or dose is determined. Depending on the type of compound, different concepts are used: for inhibitors usually the half maximal inhibitory dose (IC_50_) is used. For compounds enhancing action or mimicking BRs and thus giving a positive response, for instance increased hypocotyl elongation, the half maximal effective concentration (EC_50_) is commonly given. Instead of using different concentrations it is also possible to apply a certain dose for one individual. In that case the half maximal inhibitory dose (ID_50_) or half maximal effective dose (ED_50_) is obtained. However, the methods for determination and calculation of these parameters are the same and thus in the following, only the term IC_50_ will be used. For determination of IC_50_ a dose response curve must be established. The physiological response should be measured in a wide range of concentrations, for instance 1 nM to 100 µM, since the plot must include at least one or two points with concentrations sufficiently low that the hypocotyls almost reach the length of the control. Similarly, at the highest two to three concentrations, maximal inhibition should be reached, as indicated by no significant further decrease of the hypocotyl length with the concentration (Figure 4B and Appendix A). It is advisable using approximately half-logarithmic steps, for instance 1, 3, 10, 30 nM and so on. The obtained data can be evaluated by logit regression analysis using the following formula:
(1)$$l=min+\frac{max-min}{1+{\left(\frac{C}{{\mathrm{IC}}_{50}}\right)}^{n_{H}}}$$
where *l* is the measured hypocotyl length and *C* the concentration of the inhibitor. The further symbols are the regression parameters: the IC_50_ value; min, the minimal length observed under full inhibition; *max*, the maximal length observed with no inhibition and n_H_, the Hill coefficient.

The method of least squares usually applied for regression analysis assumes similar SDs for all values. However, this is clearly not fulfilled for hypocotyl assays used for IC_50_ determination since the standard deviation is typically higher in samples with long hypocotyls than in such with short hypocotyls, as it can be seen for instance in Figure 4B and in the other examples shown in Appendix A. This causes the problem that deviations of long hypocotyls contribute more to the sum-of-square value while that of short hypocotyls contribute less and thus the points are unequally weighed [161]. Consequently, the obtained IC_50_ values are inaccurate and in some cases it is even impossible to calculate the function since numerical approximation of the regression parameters does not converge. To circumvent these problems, data should be weighed. Evaluating many assays, we found that a simple but suitable method for weighing is 1/y. A number of statistical programs can perform such analyses but it is also possible to calculate such regressions in Excel using the solver addon. An Excel spreadsheet for logit regression with 1/y weighing is included in Appendix A and applied for four examples (IC_50_ determination for BRZ and for three replicates of voriconazole). Besides IC_50_, logit regression analysis provides also n_H_, the Hill coefficient, which indicates the number of interaction sites of the inhibitor [162]. For all triazole-type BR biosynthesis inhibitors we investigated so far, values obtained for n_H_ were close to 1, indicating the presence of one binding site. The sign of n_H_ has no meaning except that a negative sign means that the hypocotyl length decreases with the concentration, which is seen for BR biosynthesis inhibitors, while a positive sign indicates that the hypocotyl length increases with the concentration, as it is observed for instance for inhibitors of BIN2 and for compounds mimicking BL. In addition to hypocotyl elongation assays, also longitudinal growth of rice seedlings has been used as a readout [163].

For compounds acting as BR mimetic the rice lamina inclination assay can be applied. This assay was originally developed for quantification of minute amounts of bioactive BRs [164]. For this assay, rice seeds are germinated and then grown in darkness for a few days. Then segments consisting of the second leaf lamina and the second lamina joint and sheath are excised and floated on water for 24 h. Subsequently, they are transferred to water containing the compound to be tested and incubated for 48 h in the dark. Finally, the angle between the leaf and sheath is measured [164,165]. This assay has been used to compare the activities of 24-epiBL and the BL mimetics bikinin and brazide [166]. In addition, this assay is also suitable for determination of the activity of compounds inhibiting binding of BL to its receptor BRI1. In such a case, plants are co-treated with BL and the putative BL antagonist.

Another occasionally used assay for BR activity is the bean second internode test. A solution of the compound is applied to the base of the second internode of 7-day old bean (*Phaseolus vulgaris*) seedlings. After 4 days, the length of the internode is measured [89].

It must be emphasized that IC_50_ determinations are influenced by a number of factors including the species and variety, the kind of application of the drug, and the growth conditions including media, light, temperature and age of the plants. The IC_50_ for different species can vary enormously. For instance, for the BR biosynthesis inhibitor (*2S*,*4R*)-Brz220 IC_50_ values of 0.01 µM [167] and 1.21 µM [168] have been reported for cress and *A. thaliana*, respectively. Similarly, yucaizol showed IC_50_ values of 0.045 µM and 0.8 µM in *A. thaliana* and rice, respectively [163,169]. However, even for the same species and identical experimental settings, relative standard deviations (RSDs) of 30% are typically seen [170]. Thus, small differences should not be overinterpreted.

### 2.2. Target Identification

The targets of a number of compounds interfering with BR biosynthesis and signaling have been identified. Most targets of BR biosynthesis inhibitors were identified by a combination of feeding experiments and in vitro binding assays. In some cases also quantification of intermediates in BR and sterol biosynthesis was applied. Targets of inhibitors of the BR signaling machinery were mainly identified by mutant rescue experiments and enzymatic assays.

#### 2.2.1. Quantification of Precursors

Quantification of the intermediates of BR biosynthesis is, in principle, a straight forward method for identification of a blocked step. It has been used for identifying the step blocked in the *det2* mutant [51]. However, most BRs are present in plants only in minute amount, typically in the low ng/g Fw range or below. Thus, quantification is laborious and requires special equipment. Classically, BRs are extracted from the lyophilized tissue with a mixture of methanol and chloroform and deuterium-labeled internal standards are added. The extract is partitioned between water and chloroform and the organic extract is further purified by column chromatography and subsequently by preparative HPLC. Finally, the fractions are evaporated, derivatised and analyzed by gas chromatography-mass spectrometry (GC-MS) [51,171,172]. Alternatively, liquid chromatography with tandem mass spectrometry (LC-MS/MS) can be used [173]. However, many intermediates are not commercially available in pure form making preparation of standards challenging. Because of these drawbacks, this method has rarely been used. Examples for characterization of BR biosynthesis inhibitors by measuring BR profiles are BRZ [174] and YCZ-18 [73].

Measuring sterol profiles can also be very informative. For instance, voriconazole was initially thought to inhibit BR biosynthesis since the phenotype of treated plants clearly resembled BR mutants. However, by using BR and sterol profiling, it could be shown that voriconazole inhibits sterol rather than BR biosynthesis [78]. The bulk sterols campesterol, β-sitosterol and stigmasterol as well as brassicasterol in brassicaceae are present in plant tissues in the mid to high µg/g Fw range and thus quantification, for instance by GC-MS [172], is simple. If an inhibitor has more than one target site in a specific biosynthetic pathway only the first inhibited step can be identified by this technique.

#### 2.2.2. Feeding with Precursors

Rescue experiments by feeding of BR biosynthesis mutants with intermediates has been successfully applied for mutants including *dwf4* [52] and *cpd* [15]. Feeding experiments were also applied for identification of the target of several BR biosynthesis inhibitors [73,74,75,168,174,175]. For rescue experiments plants are grow in presence of the inhibitor at a concentration causing a clear phenotype, typically at the IC_50_ concentration or slightly higher. In addition, the precursors are added, usually in concentrations of 10 nM to 1 µM for BL and CS and 1 to 10 µM for the upstream precursors. No response of the hypocotyl length indicates that the applied BR intermediate is upstream of the inhibited step, while increased hypocotyl length indicates that it is downstream of the target site. However, it must be mentioned that *A. thaliana* seedlings do not show a good response to intermediates of BR biosynthesis upstream of cathasterone. This makes target site identification difficult if the BR biosynthesis acts upstream of cathasterone [176]. In addition, it must be emphasized that this technique allows only identification of the last inhibited step for compounds acting at multiple reactions.

#### 2.2.3. Mutant Rescue Experiments for BR Mimetics

For compounds activating BR signal transduction, rescue experiments with BR mutants were successfully applied [70,177]. BR biosynthesis and BR signal transduction mutants showing typical BR deficient phenotypes are grown in vitro on control media and on media supplemented with the compound to be tested. Rescue of the phenotype of BR perception mutants confirms that the compound activates BR signaling. Analysis of different BR signal transduction mutants may allow identifying the affected step. Such experiments showed that bikinin acts at the level or downstream of the kinase BIN2 and its homologues [70].

#### 2.2.4. In Vitro Enzyme Activity Assays

Targets can be verified by performing in vitro enzymatic assays. Reduction of the catalytic activity in the presence of the inhibitor is a clear sign for interaction of the compound with the enzyme. However, for CYP P450, in vitro enzymatic assays are challenging since even partial purification of bioactive CYPs is very difficult. In contrast, DET2 can be expressed in mammalian cells in its active form allowing enzymatic assays in cell hydrolysates using radio-labeled steroids as substrates. Using such assays, the inhibitory effects of a number of steroidal and non-steroidal compounds on DET2 [76,77] were determined. Similarly, BIN2 and its homologues can be expressed in *E. coli* and purified in their active forms. In vitro kinase assays using BES1 as natural substrate or myelin basic protein as an artificial substrate and [γ-^32^P]-ATP as co-substrate revealed that bikinin inhibits BIN2 and several of its homologues [70].

In addition, enzymatic assays can also be used for determination of the inhibitory constant *Ki*, which is the concentration required to produce half maximum inhibition. The velocity (*v*) of the enzymatic reaction is determined at two or more substrate concentrations and over a range of inhibitor concentrations (*I*). Plotting the reciprocal velocity (1/*v*) against the concentration of inhibitor at each concentration of substrate (the Dixon plot) gives a family of intersecting lines [178]. For competitive inhibitors the lines intersect above the *x*-axis and the concentration of *I* where they converge is -*Ki*. The same applies for non-competitive inhibitors except that they intersect at the abscissa. This method has been used to determine *Ki* of the DET2 inhibitor 4-MU [77].

#### 2.2.5. Determination of the Dissociation Constant

The dissociation constant *K_d_* is a measure for the affinity of a ligand for binding to the central molecule. A low *K_d_* means high affinity while a high *K_d_* indicates low affinity. Typical *K_d_* values of inhibitors targeting BR biosynthetic enzymes or proteins involved in signal transduction are in the high nM to low µM range. For systems with a 1:1 binding stoichiometry, meaning that one ligand molecule is bound per protein molecule, the *K_d_* is equal to the inhibitor concentration at which half of the protein molecules are bound. This stoichiometry is fulfilled for binding of triazoles to cytochrome P450 enzymes. A number of methods for determination of *K_d_* have been developed, for example measurement of the enzymatic activity in the presence of different inhibitor concentrations [179,180,181,182], isothermal calorimetry (ITC) [183,184], microscale thermophoresis [185] and surface plasmon resonance (SPR) [186,187]. Except for SPR, which was used for studying interaction of bikinin with BIN2 [70], these methods have not been applied for determination of *K_d_* constants of compounds interfering with BR biosynthesis of signaling so far.

All inhibitors of BR biosynthesis with known targets bind to cytochrome P450 enzymes. For studying such interactions, absorption spectroscopy in the visible range is highly suitable. Heme-bound iron shows strong absorption in the blue wavelength range of the visible spectrum. This absorption band is often referred to as the Soret band, named after its discoverer Jacques-Louis Soret, a 19th century Swiss chemist. The Soret band of free CYP P450 monooxygenases is typically in the range of 415–418 nm for the oxidized form and 407–409 nm for the reduced form. Upon binding, the ligand can displace a water molecule coordinated with the iron atom and shift the Soret band to approximately 385–395 nm, which is called a type I shift [188]. For instance, binding of campesterol to DWF4 caused a shift from 416 nm for the free protein to 383 nm for the sterol-bound form [61]. In contrast, binding of molecules containing unhindered nitrogen atoms, for instance triazole or imidazole derivatives, cause shifts of the Soret band to a longer wavelength, typically in the range of 420–425 nm [188]. This is called a type II shift. For instance, binding of the BR biosynthesis inhibitor (*2R*,*4S*)-BRZ220, a triazole derivative, to DWF4 caused a shift from 414 nm for the free protein to 420 nm for the ligand-bound enzyme [168].

This phenomenon can be used for determination of the dissociation constant *K_d_* by recording spectra of the CYP P450 protein in the presence of different concentrations of the inhibitor. Since the spectral shifts are small, this is done in practice by recording difference spectra using a double beam spectrophotometer. Both, the sample and the reference cuvettes, are filled with a solution containing the CYP P450 in a suitable buffer and the absorption is set to zero. Now the inhibitor is added to the sample cuvette while an equal volume of solvent is added to the reference cuvette. After equilibration (typically a few min) another spectrum is recorded. This procedure is repeated several times with increasing ligand concentrations. Importantly, the total volume added must be kept as small as possible (less than 1% of the total volume). For a compound inducing a type II shift the difference spectra will show a minimum in the low wavelength range, typically at approximately 415 nm, and a maximum in the high wavelength range, typically at approximately 435 nm. Now the difference of the maximum and minimum is calculated and plotted against the concentration of the ligand, which gives a hyperbolic function resembling a Michaelis-Menten function. *K_d_* can be calculated the using the formula shown below with Δ*ABS* being the difference of the absorption at the maximum and minimum in the difference spectrum and *C* being the concentration. Δ*ABS_max_* is the (calculated) maximal absorption difference at an infinite ligand concentration.
(2)$$\Delta ABS=\Delta ABS_{max}\times \frac{C}{C\times K_{d}}$$


For regression analysis a hyperbolic fit is used. Weighing of the data points by 1/y is possible. Alternatively, the function can be linearized by plotting 1/C on the *x*-axis and 1/Δ*ABS* on the *y*-axis. However, that approach is nowadays less frequently used since it causes weighing of the data points in an undesired way. To obtain reliable results a number of concentrations below and above the *K_d_* must be recorded. In addition, the concentration of the protein should be similar to or less than the *K_d_*; if it is (much) higher inaccurate *K_d_* values are obtained [189].

### 2.3. Specificity

BR and GA deficient mutants have a similar phenotype characterized by reduced hypocotyl and petiole elongation, dark green leaves, reduced plant height and delayed senescence. Accordingly, BR and GA biosynthesis inhibitors impact on plant growth phenotypes in a similar way. Therefore, methods have been developed for distinguishing between BR and GA biosynthesis inhibitors. The most widely used technique is co-application of the compound to be tested and BL as well as GA. The assay is performed in the same way as feeding studies with precursors described in Chapter 2.2.2. except that BL (or 24-epiBL) and a bioactive GA (usually GA_3_ but also GA_1_ has been used) are added to the medium. After incubation, the hypocotyl length of the control is compared with plants grown on medium containing the inhibitor only, the inhibitor plus BL and the inhibitor and GA. If a compound inhibits BR biosynthesis selectively, hypocotyl elongation can be restored by co-application of BL while GA has only a small impact. In contrast, if the phenotype is caused by inhibition of GA biosynthesis, co-application of BL has a minor impact while hypocotyl elongation is restored by co-application of GA. It is important to mention that BR signaling impacts on GA levels and even BR biosynthesis and perception mutants show, depending on the growth conditions, a slight to moderate response to GA application [190,191,192]. Consequently, a small response is also seen for co-application of a selective BR biosynthesis inhibitor and GA. In addition, sterol and BR biosynthesis inhibitors give the same results and thus cannot be distinguished by that method. The result of an assay using BRZ as an example for a BR biosynthesis inhibitor and voriconazole as a representative for a sterol biosynthesis inhibitor is shown in Figure 5A,B.

GA biosynthesis is essential for germination of *A. thaliana* [193,194] and compounds inhibiting its biosynthesis impede germination efficiently [195,196]. In contrast, even strong BR mutants germinate normally if plated on ½ MS medium. Reduced germination for BR mutants has only been described in the presence of abscisic acid [197]. To assay the impact on germination, *A. thaliana* seeds are plated on ½ MS medium supplemented with an appropriate concentration of the compound to be tested. As a control, seeds are plated on medium without compound. Subsequently, the plates are incubated for a few days and at 4 °C in the dark prior to incubation at 21 °C under long day conditions for a few days. Germination is usually defined by appearance of at least the radicula. Using this assay, the BR biosynthesis inhibitor BRZ2001 and the sterol biosynthesis inhibitor voriconazole did not inhibit germination while the GA biosynthesis inhibitors paclobutrazol and uniconazole inhibited germination drastically [78]. An example for such an assay using BRZ, paclobutrazol and voriconazole as representatives for BR, GA and sterol biosynthesis inhibitors, respectively, is shown in Figure 5C,D.

Most BR biosynthesis mutants known today were designed using fungicides as lead compounds. These fungicides inhibit fungal sterol synthesis by inhibiting the sterol 14-demethylase CYP51. CYP51 is the most conserved cytochrome P450 monooxygenase and homologues can be found in bacteria, fungi, yeasts, animals and plants [198]. For instance, the CYP51 protein sequence of *Penicillium italicum* causing green rot on citrus fruits during post-harvest storage shares 30% identity and 45% similarity to *A. thaliana* CYP51A2 involved in phytosterol biosynthesis. Thus, compounds inhibiting fungal CYP51 might also impact on the activity of plant CYP51 monooxygenases. As mentioned above, phenotypes caused by BR biosynthesis inhibitors and sterol biosynthesis inhibitors are very similar and, in addition, both can be rescued by application of BL or 24-epiBL. An efficient method for identification of compounds inhibiting sterol biosynthesis is determination of the sterol composition [78], for instance by GC-MS [172]. Although this method is relatively simple, it has not been used routinely for characterization of BR biosynthesis inhibitors.

Previously, it has been shown that uniconazole is an efficient inhibitor of CYP707, an P450 monooxygenase that catalyzes 8′-hydroxylation of abscisic acid [199]. In contrast, BRZ had only a minor effect. Nevertheless, abscisic catabolism might be a possible off-target for BR inhibitors. However, the assay is relatively complicated since microsomes containing active 8′-abscisic acid hydroxylases must be isolated, in vitro reactions performed in the absence and presence of the compound to be tested and phaseic acid, the product of abscisic acid hydroxylation, needs to be quantified by chromatographic techniques [199].

## 3. Inhibitors of Brassinosteroid Biosynthesis

In the last two decades a number of highly potent and specific inhibitors of BR biosynthesis have been developed. Most BR inhibitors known today belong to the triazoles [72], which is not surprising since the triazole moiety can interact with the heme-bound iron of CYP P450s and thereby block their activity [200].

### 3.1. Brassinazole and BRZ2001

First evidence for inhibition of BR biosynthesis by a triazole was obtained by treatment of *Zinnia elegans* mesophyll cells with uniconazole [11]. Uniconazole (Figure 6A) is widely used as a plant growth retardant interfering with gibberellin (GA) biosynthesis [201] by inhibiting the cytochrome P450 monooxygenase CYP701 [202,203], which catalyzes conversion of ent-kaurene to ent-kaurenoic acid [203,204]. However, the observed effect on tracheary element formation could not be explained by that activity since application of GA_3_ could not reverse the effect. In contrast, externally applied BL efficiently rescued tracheary element formation, indicating that uniconazole inhibits, in addition to GA, also BR biosynthesis [11,13]. In another study, uniconazole treatment reduced CS levels in *P. sativum*, providing additional evidence for an inhibitory effect on BR biosynthesis [205]. Despite these results, uniconazole was considered as unsuitable for manipulating BR levels since it lacks specificity. In addition to inhibiting BR and GA biosynthesis, uniconazole is also known to decrease abscisic acid (ABA) catabolism by inhibiting CYP707 [199,206], a cytochrome P450 monooxygenase hydroxylating ABA to 8′-hydroxyabscisic acid, which cyclizes likely spontaneously to phaseic acid [207,208,209,210,211]. Moreover, uniconazole also impacts on the levels of bulk phytosterols [205] and the cytokine *trans*-zeatin [212].

Min et al. used uniconazole and the structurally similar triazole paclobutrazol, another growth retardant inhibiting GA biosynthesis [213], as lead compounds for identification of specific and potent BR biosynthesis inhibitors [214]. The obtained products were first tested for inhibition of GA biosynthesis using a rice stem elongation assay since it was thought at that time that rice stem elongation depends only on GAs but not on BRs. Compounds recognized as GA inhibitors were excluded from further testing. The remaining compounds were assayed for inhibition of cress hypocotyl elongation. Two compounds, no. 6, which was later named brassinazole (BRZ) [215], and no. 10 (Figure 6B), were highly active. The phenotype of treated plants could be reversed by co-application of 10 nM BL while 1 µM GA_3_ had only a negligible effect.

The structure of compound 6 (BRZ) resembles paclobutrazol while compound 10 shares its core structure with uniconazole. However, both compounds have a phenyl residue replacing the tert-butyl group of the lead compounds. In BRZ, a methyl residue was attached to the carbon carrying the hydroxy group. Interestingly, this methyl group is essential for activity since compound 5 (Figure 6B), which is otherwise identical to BRZ, lacks any BR inhibiting activity under the conditions tested. Studies in *A. thaliana* confirmed BRZ as a potent BR biosynthesis inhibitor [215]. Although compound 10 also showed a high activity, it was, in contrast to BRZ, not further characterized.

BRZ was synthesized by bromination of acetophenone in its side chain to yield 2-bromoacetophenone, which was further reacted with 1,2,4-triazole (Figure 6B). The intermediate was alkylated with 4-chlorobenzyl chloride in the presence of sodium hydride. Finally, the obtained ketone was reacted with the Grignard reagent methylmagnesium bromide to yield BRZ. Unfortunately, no details of the reaction conditions, for instance reaction time, temperature and yield were reported [214]. Stereochemical aspects, BRZ has two chiral centers and exists therefore in four stereoisomers, were not investigated.

For identification of the BRZ target, the BR profile of *Catharanthus roseus* cells treated with 5 µM BRZ was compared to that of untreated cells. In treated cells, 6-deoxocathasterone and cathasterone and all downstream metabolites were reduced while levels of upstream metabolites were either unchanged or increased. This suggested that BRZ interferes with C-22 oxidation in BR biosynthesis [174], a step catalyzed by DWF4 [52]. To confirm binding of BRZ, DWF4 was cloned and expressed in *Escherichia coli*. The purified protein exhibited a Soret absorption band at 416 nm, which was shifted to 421 nm in the presence of BRZ. This effect was used to determine the dissociation constant *K_d_* of BRZ from DWF4, which was found to be 1.05 µM. In contrast, for uniconazole a *K_d_* of 6.9 µM was determined, showing that it has significantly lower affinity to DWF4 than BRZ [174]. Very recently, the crystal structure of DWF4 in complex with BRZ was determined by X-ray crystallography. As expected, the triazole ring was found to interact with the heme iron. Both phenyl rings interacted with apolar amino acids and the hydroxy group was involved in a hydrogen bond [216]. According to that data, the bound BRZ version was the (*2R*,*3S*)-BRZ enantiomer, indicating that this stereoisomer might be the most active form. In a number of previous publications, the (*2S*,*3R*)-BRZ stereomer was shown, although the racemic mixture was used [174,176,200,215,217]. Recently, a method for enantioselective synthesis of (*2R*,*3R*)-BRZ and (*2R*,*3S*)-BRZ has been developed [218]. Both compounds were obtained with a purity of 92% ee (enantiomer excess). However, their biological activities were not investigated. Thus, a detailed stereochemical structure-activity relationship study would be helpful to verify which BRZ stereoisomer shows the highest biological activity. Importantly, whenever BRZ is applied, it should be reported whether the racemic mixture or a specific enantiomer was used. Also binding of uniconazole to DWF4 was investigated by X-ray crystallography [216]. Although it has a structure similar to BRZ, the results indicated that its binding conformation was quite different.

BRZ2001 is a variant of BRZ containing an allyl moiety instead of the methyl group (Figure 6A). The synthesis of BRZ2001 was not explicitly reported but it can be assumed that it was prepared in the same way as BRZ except that in the last step allylmagnesium bromide was used instead of methylmagnesium bromide (Figure 6B). BRZ2001 has approximately the same potency like BRZ in the *A. thaliana* hypocotyl elongation assay with IC_50_ values of 0.55 µM and 0.58 µM, respectively [219]. BRZ2001 is thought to be more specific than BRZ since BRZ-treated cress seedlings reacted more strongly to GA co-application than seedlings treated with BRZ2001 [220]. Binding of BRZ2001 to DWF4 was confirmed using a quartz-microbalance sensor [221], showing that it has the same target like BRZ. In rice, neither BRZ nor BRZ2001 could retard rice seedling elongation even at concentrations of 10 µM [220] indicating that both compounds lack activity in rice and probably other monocots.

Since its discovery, BRZ has become the most widely applied BR biosynthesis inhibitor. So far, BRZ has been applied in hundreds of studies, for instance for the isolation of the *A. thaliana* mutants *bzr1-1D* and *bes1-D* whose effected proteins act as transcription factors in BR signal transduction [69], for studying the role of reactive oxygen in BR-induced stress tolerance in cucumber [222] or for investigating the role of BRs in root nodule formation in soy bean (*Glycine max*) [223]. Factors contributing to its popularity include its high activity with IC_50_ values of less than 1 µM, its applicability to many plant species and its high specificity. For instance, it does not directly interfere with GA biosynthesis [214] and has only a minimal inhibitory effect on ABA catabolism [199]. In addition, it is commercially available, however, at a considerable price. A drawback of BRZ is its low activity in monocots. Since BRZ2001 is not commercially available, its application has remained very limited.

### 3.2. Brz220

Development of Brz220 was initiated by the observation that propiconazole (Figure 7A), a fungicide widely used in agriculture, inhibits BR biosynthesis [167]. A number of compounds were synthesized by bromination of substituted acetophenones to 2′-bromoacetophenones, which were acetalyzed with 1,2-dihydroxypentane to obtain the dioxolane ring [224]. The obtained products were reacted with 1,2,4-triazole to yield the final product. Figure 7B shows the reaction sequence for the synthesis of Brz220. The biological activity of the derivatives was determined by using the cress hypocotyl elongation assay. Co-application of BL and GA_3_ was used to investigate specificity for inhibition of BR biosynthesis. Compound 12 [167], which was later named Brz220 [224], was identified as the most active compound and showed high specificity for inhibition of BR rather than GA biosynthesis.

Like BRZ, Brz220 contains two stereocenters and therefore exists in four stereoisomers. To determine the stereochemical structure-activity relationship, the diastereomers were separated by silica gel chromatography using ethyl acetate/hexane (85/15) as eluent. The enantiomers were separated by semi-preparative chiral HPLC using a Chiralpak AS column and hexane/2-propanol (9:1) as eluent. The (*2S*,*4R*)-Brz220 stereoisomer (also named Brz22012) showed by far highest activity with an IC_50_ of 0.01 µM in the cress hypocotyl elongation assay [224].

Co-treatment of *A. thaliana* plants with Brz220 stereoisomers and different BRs showed that all tested compounds including cathasterone, teasterone, typhasterol, CS and BL rescued stunted the growth to some extent. The same results were obtained with all four stereoisomers indicating that they target the same site. More importantly, this experiment suggested that Brz220 inhibits a reaction step upstream of the conversion of cathasterone to teasterone. DWF4, likely catalyzing conversion of 6-oxocampestanol to cathasterone, was considered as the most likely candidate. To investigate this in more detail, the *A. thaliana* DWF4 gene was cloned and heterologously expressed in *E. coli*. The free protein showed a Soret absorption band with a maximum at 414 nm, which was shifted to 420 nm by addition of the Brz220 stereoisomers. Titration of the protein with the compounds allowed determination of the dissociation constant *K_d_*, which was found to be 0.22 µM for (*2S*,*4R*)-Brz220, the most active stereoisomer [224]. This study was the first to show the strong impact of stereochemical properties on the activity of a BR biosynthesis inhibitor.

### 3.3. The YCZ Series

Another series of BR biosynthesis inhibitors possessing a dioxolane ring, in addition to the triazole ring, were designed using ketoconazole (Figure 8A) as a lead compound. Ketoconazole is used in medicine [225] and veterinary medicine [226,227] for treatment of fungal infections, mainly by application on the skin [228,229,230] but in some cases also administrated orally [231]. Ketoconazole inhibits fungal ergosterol biosynthesis by binding to CYP51 [232], a cytochrome P450 14α-demethylase catalyzing oxidative demethylation of lanosterol [233].

Ketoconazole and Brz220 share the dioxazole ring and a substituted phenyl at C-2. Oh and co-workers decided to synthesize a number of compounds with different phenoxy substituents [170]. Synthesis started with reaction of 4-chlorophenacyl bromide with 1,2,4-triazole. The obtained product was acetalized with 1-tosylglycerol, which had been prepared from commercially available 1,2-isopropylidene glycerol. Finally, the tosyl group of the obtained intermediate was substituted with phenolic compounds under alkaline conditions to introduce the desired phenoxy group (Figure 8B). The obtained versions were tested using the *A. thaliana* hypocotyl elongation assay and the IC_50_ concentrations were determined. The most active compound, called 7m in this study and YCZ-18 in subsequent manuscripts (Figure 8A), contained a 2-ethoxyphenyl residue and showed an IC_50_ of 0.10 ± 0.03 µM. This identified it as approximately 6-times more active than BRZ, which showed an IC_50_ of 0.58 ± 0.38 µM in the same assay. In addition, the specificity for inhibition of BR biosynthesis was also investigated by co-application of 10 nM BL or 1 µM GA. Only BL rescued hypocotyl elongation almost to a length of untreated plants while GA application had minute effects, confirming that compound 7m/YCZ-18 specifically inhibits BR biosynthesis [170]. In a subsequent study, Oh and co-workers continued the SAR (structure-activity relationship) study by testing 15 additional modifications of the phenoxy-group [169]. Based on previous results, mainly derivatives with an ortho-substitution were investigated. YCZ-18 was used as a reference and showed an IC_50_ of 0.13 ± 0.01 µM. Only two compounds, G1 (called yucaizole in a subsequent study [163]) containing a 2-propoxyphenyl residue and G2 possessing a 2-allyloxyphenyl moiety (Figure 8A), showed higher activities than YCZ-18 with IC_50_ values of 0.06 ± 0.01 µM and 0.05 ± 0.01 µM, respectively. 

Again, the specificity for inhibition of BR biosynthesis was confirmed by co-application of BL and GA. The results indicated that bulky residues like a napthyl or a 2-benzylphenyl residue reduced the activity significantly. Later on, that information was used to design a fluorescent YCZ derivative by incorporating a 6-hydroxycoumarine residue into the molecule. The obtained compound was designated YCZ-FL (Figure 8A) and exhibited pronounced fluorescence with an excitation maximum at 336 nm and an emission maximum at 442 nm. The IC_50_ of the compound was 1.56 µM [234], which is considerably higher than that of YCZ-18. Nevertheless, its binding strength might be sufficient for binding studies.

To establish the role of the triazole ring for biological activity, a number of compounds containing other substituents were synthesized. Replacing the triazole ring by an imidazole moiety reduced the activity by approximately 7-fold, and replacing it by a pyrazole ring led to an almost inactive derivative. This confirmed that the triazole group is essential for high inhibitory activity [169].

To this end, the phenoxy group was optimized and the relevance of the triazole ring confirmed. In order to study the impact of the 2-phenyl group, eight derivatives with different residues were synthesized using the established route [235] (Figure 8B). In this study, a 2-trifluorophenoxy residue was used as substituent for C-5. The compound with the highest activity was called YCZ-14 (Figure 8B) and showed an IC_50_ value of 0.12 ± 0.04 µM in the *A. thaliana* hypocotyl elongation assay. Feeding with BL or GA confirmed its specificity for inhibition of BR biosynthesis. According to these data, the C-2 substituent of YCZ-18 and yucaizole is optimal since it is identical to that of YCZ-14.

A selection of the previously synthesized compounds was tested in rice as a model for monocots. In a first study [219], compounds containing the 4-chlorophenyl group at C-2 and a phenoxy group with or without chlorine substituents were tested. Among them, compound 1 (formerly called compound 7a [170]; Figure 8A) containing a non-substituted phenoxy moiety showed highest activity with an IC_50_ of 1.27 ± 0.43 µM for retardation of rice seedling elongation. The chlorine substituents decreased the activity. Among the tested substitution patterns a single chlorine in position 2 had the least inactivating effect. In a second study [163] also yucaizole, one of the most active compounds identified using the *A. thaliana* hypocotyl assay, was tested in rice. Indeed, it showed with an IC_50_ of 0.8 µM for retardation of rice seedling elongation a slightly higher activity than compound 1, confirming that a 2-propoxy substituent enhances activity.

It is important to mention that the compounds of the YCZ series, like BRZ, BRZ2001 and Brz220, contain two stereocenters and therefore exist in four stereoisomers. To investigate the impact of the stereochemical configuration on biological activity, pure stereoisomers were prepared. The structure was based on G2, the most active compound identified, except that the phenyl ring placed on C-2 was 2,4-dichloro substituted. The obtained product was called YCZ-2013 [236]. Recently, the stereochemical properties of an intermediate in itraconazole synthesis were studied in considerable detail [237]. Selecting a 2,4-dichloro substitution pattern for synthesis of YCZ-2013 lead to the same intermediate as for itrraconazole synthesis, which facilitated determination of the absolute configuration of YCZ-2013. The mixture of the diastereomers (*2R*,*4S*)-YCZ-2013 and (*2S*,*4S*)-YCZ-2013 was obtained by using enantiopure (*R*)-isopropylideneglycerol for the general synthetic route (Figure 8B). The obtained diastereomers were separated by recrystallization and preparative thin layer chromatography and further derivatized with 2-(allyloxy)phenol to (*2R*,*4R*)-YCZ2013 and (*2S*,*4R*)-YCZ-2013. The two remaining stereoisomers, (*2S*,*4S*)-YCZ-2013 and (*2R*,*4S*)-YCZ-2013, were obtained using enantiopure (*S*)-isopropylideneglycerol for their synthesis. Interestingly, both, the (*2R*,*4S*)-YCZ-2013 and (*2S*,*4R*)-YCZ-2013 enantiomers were highly active in *A. thaliana* hypocotyl elongation assays with IC_50_ values of 0.024 ± 0.002 µM and 0.024 ± 0.003 µM, respectively. This result was unexpected since in case of structurally closely related Brz220 only (*2S*,*4R*)-Brz220 was highly active with an IC_50_ of 0.01 µM in cress hypocotyl elongation assays while its enantiomer (*2R*,*4S*)-Brz220 was markedly less active with an IC_50_ of 0.12 µM [224].

For identification of the target of YCZ inhibitors, YCZ-18 was selected. To narrow down the affected biosynthetic step, rescue of the phenotype of YCZ-18-treated plants was assayed with different BRs. Campestanol [169] and cathasterone could not rescue the phenotype while teasterone restored hypocotyl elongation of dark grown *A. thaliana* seedlings [73]. This pointed to inhibition of the conversion of cathasterone to teasterone. This reaction is catalyzed by the two closely related enzymes ROT3 and CYP90D1. Heterologous protein expression in *E. coli* failed for ROT3, but sufficient CYP90D1 could be obtained for performing in vitro binding assays. Like for other triazole inhibitors, the spectral shift of the Soret absorption band after addition of the inhibitor was used as a readout. A shift from 421 nm to 425 nm was observed after addition of YCZ-18. Titration of the protein with the inhibitor revealed a dissociation constant *K_d_* of 0.79 µM. The result that YCZ-18 targets CYP90D1 and likely also ROT3 is surprising since binding to DWF4 had previously been shown for the structurally closely related inhibitor Brz220 [168]. In addition, a BR profile of YCZ-18 treated plants was not supportive for inhibition of CYP90D1 and ROT3 since the level of 6-deoxoteasterone was not reduced in treated as compared to control plants [73]. Thus, it cannot be excluded that the YCZ inhibitors might also have a second target site. In any case, YCZ-18, yucaizole and the (*2R*,*4S*)/(*2S*,*4R*)-YCZ-2013 stereoisomers are the most potent BR biosynthesis inhibitors known today. In contrast to BRZ and BRZ2001, they are also highly active in rice [163,219] and probably other monocots.

### 3.4. Propiconazole

Propiconazole (Figure 7A) is used in agriculture for treatment of infections of crops by a broad range of fungi including *Erysiphe graminis* [238,239,240], *Septoria* spp. [240,241,242,243], *Puccinia* spp. [244,245,246] and *Fusarium* spp. [247,248,249]. In fungi, propiconazole inhibits lanosterol 14α-demethylase (CYP51) [250,251]. If inhibited, ergosterol, the bulk sterol of fungi, cannot be synthesized. In contrast, a number of unusual, partly toxic, sterols are formed, which inhibits fungal growth. Propiconazole can also inhibit CYP51 of *Candida albicans*, human [252] and zebrafish [253]. As mentioned above, propiconazole was used as a lead compound for designing Brz220. However, Brz220 is not commercially available and BRZ is quite expensive. In addition, the latter is inactive in rice and probably other monocots [219,220]. Thus, the idea of using propiconazole directly was found attractive since it is readily available at a low price, which makes it interesting particularly for application in agriculture and horticulture crop species.

First evidence for an inhibition of BR biosynthesis by propiconazole was obtained in cress hypocotyl elongation assays, where a pronounced shortening of the hypocotyl by application of 1 µM propiconazole was observed. Application of 10 nM BL reverted hypocotyl length to untreated values while 1 µM GA_3_ had a minute effect, confirming that BR but not GA biosynthesis is targeted [167]. Inhibition of BR biosynthesis was also observed in *A. thaliana* and in *Zea mais* [213,254]. For the latter a strong dependency on the genetic background was observed [254]. In addition, application of propiconazole increased the mRNA level of *CPD*, *DWF4* and particularly *BR6ox2* in *A. thaliana*, a reaction typically seen in BR-deficient plants [254].

Feeding of propiconazole-treated *A. thaliana* plants with BRs showed that the phenotype could be rescued by teasterone, CS and BL but not by campestanol and campesterol, indicating that either C-22 hydroxylation catalyzed by DWF4 or the subsequent step, C-23 hydroxylation catalyzed by ROT3 and CYP90D1, is inhibited. Expression of DWF4 and ROT3 failed while CYP90D1 was successfully expressed in *E. coli*. Thus, binding assays could only be performed with the latter protein. Free CYP90D1 showed a Soret band with a maximum of 414 nm, which shifted to 437 nm upon binding of propiconazole. Titration revealed a dissociation constant *K_d_* of 0.76 µM [75].

Based on these studies, propiconazole is now increasingly used for inhibition of BR biosynthesis. However, it must be kept in mind that propiconazole is known to inhibit CYP51, a crucial enzyme for synthesis of bulk sterols, in fungi and mammals, and so far the impact of propiconazole on plant sterol synthesis has not been investigated. Thus, based on the available data it is currently not possible to exclude that propiconazole, in addition to CYP90D1, inhibits also CYP51. In the future, studies should be performed to investigate the specificity of propiconazole in more detail.

### 3.5. Triadimefon

Triadimefon (Figure 9A) is a triazole-type fungicide used in agriculture for treatment of infections by *Erysiphe graminis* the causal agent of powdery mildew [255,256,257], *Septoria* spp., the causative agent of leaf spot diseases [258], and diverse fungi causing rusts [259]. Similar to propiconazole, triadimefon inhibits oxidative 14α-demethylation in fungal ergosterol biosynthesis. Plants treated with triadimefon show clear growth responses including reduced elongation of stems and petioles, dark green and thicker leaves, and delayed senescence [260]. Particularly for ornamental plants, these side effects are considered beneficial since treated plants look more attractive, which increases their market value [261]. In addition, triadimefon treatment inhibits GA and sterol biosynthesis in plants to some extent [262] and stimulates production of cytokinins and chlorophyll [260].

Phenotypes of both light and dark-grown *A. thaliana* plants treated with triadimefon resembled BR biosynthesis mutants [175] like *det2* [51] and *cpd* [15]. These phenotypes could be partially rescued by application of BL, indicating that triadimefon might inhibit BR biosynthesis. In addition, triadimefon-treated plants showed increased *CPD* expression levels [175]. Feeding with different intermediates of BR biosynthesis showed that the phenotypes could be rescued with cathasterone, teasterone, typhasterol, CS and BL, suggesting a block in BR biosynthesis upstream of cathasterone [175]. These results resembled BRZ treatment [174] and suggested that DWF4 might be a possible target. Indeed, binding studies with DWF4 revealed a dissociation constant *K_d_* of 2.5 µM, which is approximately 2.5 times higher than that of BRZ [175]. This indicated that triadimefon has a lower affinity than BRZ. In line with that result, hypocotyl elongation assays revealed an IC_50_ in the range of approximately 1.5 µM, which is approximately 3 times higher than that of BRZ. Rescue experiments with BL and GA_3_ using cress seedlings showed that BRZ-treated seedlings were fully rescued by BL (105% hypocotyl length of the control) but minimally by GA_3_ (28% length of the control). In contrast, triadimefon-treated seedlings were rescued with BL to 82% length of the control and with GA_3_ to some extent (48% length of the control). Taken together, these data indicate that triadimefon targets BR biosynthesis but with a lower efficiency than BRZ. Moreover, it is less specific since triadimefon inhibits GA biosynthesis to a much larger extent than BRZ does. Triadimefon has not been used in research for inhibiting BR biosynthesis since a more potent and specific inhibitor is available with BRZ. However, it might still be useful for production of more attractive ornamental plants.

### 3.6. Imazalil

Imazalil (Figure 9A) is an imidazole fungicide mainly used as an antifungal post-harvest agent, particularly for preventing growth of *Penicillium digitatum* and *P. italicum* on citrus fruits during storage [263,264,265,266]. Like the triazole fungicides triadimefon and propiconazole, imazalil inhibits fungal biosynthesis of ergosterol [267,268] by binding to CYP51 [269].

Imazalil caused severe hypocotyl shortening in *A. thaliana*, which could be reversed largely by application of 24-epibrassinolide (EBL) and fully by co-application of EBL and GA_3_. In contrast, GA_3_ alone had little effect [217]. The hypocotyl length of the BR deficient mutant *dwf4* was not further shortened by imazalil while *etr1-3*, which cannot sense ethylene, responded to imazalil.

Cell length was clearly reduced by imazalil treatment and the *CPD* mRNA level was found to be increased. The IC_50_ in an *A. thaliana* hypocotyl elongation assay with plants grown in the darkness was 3.44 µM, while BRZ and BRZ2001 showed IC_50_ values of 0.77 µM and 0.33 µM, respectively. Thus, imazalil was 5-times less effective than BRZ and 10-times less active than BRZ2001 [217]. These data suggest that imazalil inhibits BR biosynthesis although with a lower potency and specificity than BRZ. The target of imazalil remains elusive.

### 3.7. Fenarimol and DDP4

Fenarimol (Figure 9A) is a pyrimidine fungicide used mainly for protection of ornamental plants, fruit trees, grapes and vegetables against powdery mildew [270], scab [271,272,273] and rusts [274]. Pyrimidine derivatives like fenarimol are known to inhibit cytochrome P450 monooxidases involved in 14α-demethylation in fungal ergosterol biosynthesis [275], ent-kaurene oxidation crucial for GA biosynthesis in plants [276], and aromatase required in mammals for estrogen synthesis [277].

Fenarimol was found to phenocopy BR-deficient mutants with short hypocotyls, deetiolation of dark grown seedlings and dark green, downward curled leaves of light grown *A. thaliana* plants [74,278]. A number of fenarimol derivatives were prepared by reaction of 5-bromopyrimidine and substituted benzophenones in the presence of n-butyllithium (Figure 9B). The obtained products were assayed for biological activity using the cress hypocotyl elongation assay, which identified DPPM4 (Figure 9B) as the most active compound. The phenotype of shortened hypocotyls could be rescued by application of 10 nM BL but not by 1 µM GA, indicating specificity for targeting BR rather than GA biosynthesis [278].

Since the structure of DPPM4 and fenarimol are highly similar (Figure 7), commercially available fenarimol was selected for further characterization in a follow-up study. The IC_50_ of fenarimol was 1.8 ± 0.2 µM in the *A. thaliana* hypocotyl elongation assay using dark-grown seedlings [74]. In addition, BR marker genes including *BR6ox2*, *DWF4* and *ROT3* showed increased expression upon treatment of *A. thaliana* seedlings with fenarimol. Co-application of fenarimol with different BRs showed that the shortened hypocotyl phenotype of dark-grown *A. thaliana* seedlings could be rescued by teasterone, CS and BL but not by campesterol and campestanol. These data suggested that DWF4, ROT3 and/or CYP90D1 might be inhibited by fenarimol. Among these candidates, binding to CYP90D1 was investigated by in vitro assays. The Soret band of free CYP90D1 showed an absorption maximum at 421 nm, which was shifted to 426 nm upon addition of fenarimol. Using this effect, titration of the protein with the compound revealed a dissociation constant *K_d_* of 0.79 µM for fenarimol [74]. Since DWF4 and ROT3 were not tested it remains elusive whether fenarimol might also bind to these proteins.

### 3.8. Steroidal and Non-Steroidal Inhibitors of Steroid 5α-Reductase Activity

In humans, reduction of the Δ^4,5^ double bond of a variety of steroids including testosterone, progesterone and cortisol is catalyzed by the three isoenzymes of steroid 5α-reductase, SRD5A1 [279], SRD5A2 [280], and SRD5A3 [281]. Inhibitors of these enzymes are used in human therapy for treatment of enlarged prostate and scalp hair loss. They have antiandrogenic effects by preventing conversion of testosterone to the more potent androgen dihydrotestosterone.

Cloning of the *A. thaliana DET2* gene revealed that the encoded protein shares significant homology to human steroid 5α-reductases [282]. Subsequent in vitro experiments confirmed that DET2 has steroid 5α-reductase activity and that it can be inhibited by 4-MA (Figure 10A), an inhibitor of mammalian steroid 5α-reductases [77]. The inhibition constant *K_i_* of 4-MA for DET2 was in the range of 300 nM while inhibition constants of 4-MA for SRD5A1 and SRD5A2 were reported to be 4 nM and 8 nM, respectively. Thus, although 4-MA is more potent in inhibiting the two human steroid 5α-reductases, it is still an effective DET2 inhibitor. Using tomato DET2, a number of further steroidal and also several non-steroidal compounds were tested for inhibition of plant 5α-reductases [76]. These experiments showed that the steroidal compounds 4-MA, VG106, DSMEN21 and, although at a lower potency, the commercially available drug finastride (Figure 10A) could inhibit DET2 activity in vitro. Also the non-steroidal compound AFA76 (Figure 10B) proofed to be an efficient DET2 inhibitor. Tomato seedlings grown in the presence of 20 µM AFA76 showed reduced hypocotyl length (42% reduction in dark grown plants and 25% in light grown seedlings) and shortened hypocotyls.

Sterol 5α-reductase inhibitors are clearly interesting tools for BR research. However, compared to triazoles, they are poorly characterized since IC_50_ values have not been reported so far and their activity at the molecular level, for instance on gene expression, has not been investigated. In addition, only finastride, which is one of the least active 5α-reductase inhibitors in plants [76], is commercially available. So far, finastride has been used to investigate the role of DET2 in fiber cell initiation and elongation in cotton [283] and for inhibiting BR biosynthesis in grapevine [284].

### 3.9. Spironolactone

Asami and co-workers screened steroid derivatives for phenotypes resembling BR mutants and identified spironolactone as a promising candidate [285]. Spironolactone (Figure 10C) is a drug used for treatment of heart failure [286], high blood pressure [287,288,289] and low blood potassium [290,291]. It reduces aldosterone levels and has antiandrogenic activity [292]. Spironolactone is known to inhibit both the aldosterone receptor [293] and 17β-hydroxysteroid dehydrogenase [294,295,296].

*A. thaliana* seedlings grown on medium supplemented with 10–100 µM spironolactone showed dark, downward curled leaves and shortened hypocotyls, which could be reverted by application of BL. Cress hypocotyls were also shortened, and could be rescued by BL but not GA_1_. Rescue experiments with BR biosynthesis intermediates did not give a clear result but showed that spironolactone targets an enzyme acting upstream of 6-deoxocathasterone [285].

A clear drawback of spironolactone is its low potency: a concentration of approximately 100 µM is required to obtain a phenotype similar to BR deficient plants. This might explain its limited application for inhibiting BR biosynthesis in plants although it is commercially available.

### 3.10. Brassinopride

An *A. thaliana* line transgenic for the *E. coli* glucuronidase (*GUS*) gene under control of the *CPD* promoter [297] was used for screening a chemical library containing 10,000 compounds [298]. Seedlings with shortened hypocotyls were GUS stained and such showing an induction of *CPD* promoter activity were considered potential candidates. After retesting, one compound designated brassinopride (Figure 11), was identified as a BR biosynthesis inhibitor. Hypocotyl elongation assays using dark grown *A. thaliana* revealed an IC_50_ of 17 µM. The phenotype of brassinopride-treated seedlings could be rescued by co-application of BL but not by GA, demonstrating specificity for inhibition of BR rather than GA biosynthesis. Interestingly, while dark-grown BR mutants like *cpd* [15], *det2-1* [18,299] and *cbb1* [299] as well as BRZ-treated plants [215] are known to lack apical hook formation, treatment with brassinopride enhanced apical hook formation [298]. Treatment of the ethylene insensitive mutant *ein2-2* with brassinopride reduced hypocotyl length but did not enhance apical hook formation. Similarly, silver nitrate, an ethylene perception inhibitor, prevented exaggerated apical hook formation in brassinopride-treated plants. These data indicate that brassinopride reduces BR biosynthesis while it increases ethylene production. However, it is not clear whether these processes are directly or indirectly targeted.

Analysis of compounds with similar structures to brassinopride revealed that the cyclopropyl ring is not essential since compound a3 (Figure 11) showed a similar activity for hypocotyl shortening and apical hook induction like brassinopride. Also plants treated with compounds a6 and a9 (Figure 11) showed enhanced apical hook formation to a similar extent like brassinopride while they were less active in reducing hypocotyl length. All other tested compounds were inactive in both assays [298].

The relatively low activity with an IC_50_ of 17 µM and presence of a second target affecting ethylene biosynthesis are drawbacks of brassinopride. However, brassinopride is an amide rather than a triazole or an imidazole, suggesting that it might target a BR biosynthetic enzyme distinct from a cytochrome CYP P450 monooxygenase. Thus, identification of its target in BR biosynthesis would clearly be of high interest.

### 3.11. Voriconazole Impacts on BR Levels via Inhibition of Sterol Biosynthesis

In a screen for pharmaceuticals altering BR homeostasis in *A. thaliana*, fluconazole was found to induce a typical BR-deficient phenotype characterized by dark green, epinastic leaves, shortened hypocotyls and a reduced overall size [78]. Since the potency of fluconazole was relatively low, structurally related drugs were analyzed and voriconazole was found to induce similar phenotypes already at very low concentrations. Voriconazole (Figure 12) is enantiopure (*2R*,*3S*)-2-(2,4-difluorophenyl)-3-(5-fluoro-4-pyrimidinyl)-1-(1*H*-1,2,4-triazol-1-yl)-2-butanole and is used as a therapeutic drug for treatment of fungal infections, particularly invasive aspergillosis and candidiasis [300]. Voriconazole combines a triazole moiety and a pyrimidine residue. It blocks fungal ergosterol biosynthesis by inhibiting CYP51 [301,302].

Using the cress hypocotyl elongation assay an IC_50_ of 0.083 ± 0.019 µM was determined for voriconazole (Appendix A) indicating that it has a high potency. Reduced hypocotyl length could be rescued in *A. thaliana* by application of 24-epiBL but not by GA_3_. In addition, while the GA biosynthesis inhibitors paclobutrazol and uniconazole inhibited germination of *A. thaliana* significantly, voriconazole, fluconazole and the BR biosynthesis inhibitor BRZ2001 had no impact on germination of *A. thaliana* [78]. These data indicated that voriconazole inhibits BR rather than GA biosynthesis. Uptake experiments showed that voriconazole entered the tissue of *A. thaliana* rapidly and reached a plateau level within 3 h, which was maintained for at least 96 h, indicating that voriconazole is not or only at a very low rate modified or degraded in plant tissues.

To identify the target of voriconazole, BR and sterol profiles of voriconazole-treated seedlings were compared to control plants. Interestingly, all detected BRs but also all sterols downstream of obtusifoliol were present at strongly reduced levels. This provided evidence that CYP51, a cytochrome P450 crucial for phytosterol biosynthesis, is the primary target rather than an enzyme involved in BR-biosynthesis is inhibited by voriconazole. Active *A. thaliana* CYP51 could not be expressed heterologously in *E. coli* impeding binding assays using the shift of the Soret band as a readout. Thus, another way was used for confirmation that plant CYP51 is targeted by voriconazole: in a sensitivity test a large range of plant species including rice, maize, tomato, spinach, *Zinnia elegans*, tobacco, pea, cucumber, cotton and rapeseed were tested and were clearly inhibited by voriconazole. Only woodland strawberry (*Fragaria vesca*) showed high resistance against the drug. Cloning of *F. vesca* CYP51 (*FvCYP51*) and expression in *A. thaliana* resulted in transgenic plants with clearly increased resistance against voriconazole, providing evidence that CYP51 is the primary target [78]. Thus, voriconazole lowers BR levels indirectly by decreasing the level of the bulk sterol campesterol, the precursor of BR biosynthesis.

It would be interesting to address whether other inhibitors of plant sterol biosynthesis can induce similar phenotypes like voriconazole. For instance, fenpropimorph (Figure 12) inhibiting cycloeucalenol-obtusifoliol isomerase, the step directly upstream of that catalyzed by CYP51 [303], reduces the levels of bulk sterols significantly [304]. Thus, also fenpropimorph might reduce BR levels.

## 4. Compounds Impacting on Brassinosteroid Signaling

To date several dozen of natural BRs are known with BL and CS being the biologically most active ones [72]. In addition, many synthetic BR analogs have been developed, which can be used as less expensive bioactive replacements for BL [305,306,307]. Also fluorescence labeled BR analogs have been developed, which have been useful for investigating endocytosis and endomembrane trafficking of the BR receptor BRI1 [308,309]. However, the aforementioned compounds act like BL and have been discussed in a number of excellent publications [72,305]. They will not be a topic of this review. In contrast, we will focus on small molecules inhibiting compounds of the BR signal transduction machinery. Such substances can be classified into three groups: steroidal compounds inhibiting BR perception, non-steroidal compounds inhibiting or activating the BRI1/BAK1 receptor and small molecules interfering with BR signal transduction. The latter group consists of a number of compounds inhibiting the protein kinase BIN2 and its homologues. In addition, there are two compounds, KM-01 and F1874-108 with unknown targets.

### 4.1. Brassinolide-2,3-Acetonide: A Steroidal Inhibitor of BR Perception

The crystal structure of the extracellular leucine-rich repeat domain BRI1 in complex with BL [109,110] showed that the aliphatic side chain and the C and D rings of BL (Figure 13) integrate into a pocket of BRI1. The 22-hydroxy group interacts with Tyr597 and the 23-hydroxy group interacts with Tyr646 and Ser647. The remaining parts of the A and B rings are solvent exposed and are likely required for binding the *N*-terminal part of the co-receptor BAK1 (or SERK1). This indicates that a compound possessing the 22- and 23-hydroxy groups but containing modifications of the 2- and 3-hydroxy groups might be able to bind BRI1 but disturb interaction of BRI1 with BAK1 or SERK1. Such a compound would compete with BL for binding to BRI1 and thus act as a competitive inhibitor. To test that hypothesis, Muto and Todoroki synthesized a number of BL derivatives with modification on the hydroxy groups by acetalization with acetone and other ketones [310]. Rice lamina inclination assays confirmed that none of the compounds had BR activity. Co-application of BL and the derivatives showed that brassinolide-2,3-acetonide (Figure 13) inhibited BL activity as expected. Surprisingly, also brassinolide-22,23-acetonide (Figure 13) showed an inhibitory effect although at a much lower potency than brassinolide-2,3-acetonide. BL diacetonide having modifications on both diols had no activity, probably because of its too low polarity preventing interaction with BRI1 or uptake into the cell. These findings provide additional evidence for a role of the 2- and 3-hydroxy groups in interaction with the co-receptor and show a way for designing small molecules preventing protein-protein interactions in BR perception.

### 4.2. Non-Steroidal Compounds Inhibiting or Activating BRI1

Non-steroidal compounds activating or inhibiting BRI1 are highly interesting since they are promising tools for regulating BR signaling. All attempts for designing such compounds were based on in silico modeling of molecules that might mimic the structure of BL and docking predictions into the BRI1/SERK1 complex. Indeed, recently several compounds inhibiting binding of BL to its receptor BRI1 and one compound acting as BL mimetic could be identified. Activities of several compounds could be confirmed in bioassays.

#### 4.2.1. (*E*)-1,2-Bis[*trans*-(4aα,8aβ)-4-Oxo-6α,7α-dihydroxy-4a,5,6,7,8,8a-hexahydro-(3*H*)-naphthyl]-ethylene

This compound (Figure 14) was the first reported non-steroidal mimetic of BL [311]. The design of the compound was based on data from structure-activity-relationship studies and modeling of the structure of BL and the designed mimetics. In total, 12 compounds were synthesized and assayed using the rice lamina inclination test. None of the compounds showed activity on its own. If co-applied with indole-3-acetic acid, some compounds showed activity. Among them, (*E*)-1,2-bis[*trans*-(4aα,8aβ)-4-oxo-6α,7α-dihydroxy-4a,5,6,7,8,8a-hexahydro-(3*H*)-naphthyl]-ethylene was most active. Doses as low as 1 pg/plant were reported to be sufficient for inducing a strong response in the rice lamina inclination assay. However, the dose-response curves shown seem illogical since the strength of the response was independent of the dose of the mimetic. Indeed, the compound was synthesized by another group and it was found to be biologically inactive against cress treated with BRZ [312]. Thus, the originally reported high activity could not be confirmed.

#### 4.2.2. Phenylfuran Derivatives

The availability of the structure of the extracellular domain of BRI1 in complex with BL [109,110] initiated not only the design of the steroidal BL antagonist brassinolide-2,3-acetonide (see Chapter 4.1) but enabled also the search for non-steroidal antagonists and agonists. Pharmacophore models were generated by different bioinformatic approaches and several potential candidates were identified. In one study, 22 candidate compounds were identified as potential BL antagonists, 15 of which were assayed experimentally while no putative BL mimetic could be identified [313]. Three of the 15 putative antagonists showed indeed inhibition of BL-induced rice lamina inclination. Two of them, compound 1 and 2 (Figure 15), were phenylfuran derivatives with half-maximal inhibitory doses (ID_50_) of 5.0 and 3.2 nmol/plant. Interestingly, in another study two further phenylfuran derivatives, compound 4 and 6 (Figure 15) were identified, which acted, in contrast to the perviously identified compounds as BL mimetics, as determined by radish hypocotyl elongation bioassays. Both compounds showed an activity similar of that of BL [314]. However, compound 6 inhibited hypocotyl elongation at a concentration of 100 nM, indicating that it might be toxic at elevated levels. For none of the compounds synthesis has been reported.

#### 4.2.3. Benzene-1,4-Diyldimethanediyl Di-3-Ethyl-5-Methyl-1,2-Oxazole-4-Carboxylate

Besides identification of two phenylfuran derivatives, Takimoto et al. also reported a 1,4-benzenedimethanol derivative (compound 14; Figure 16) as BL antagonist [313]. This compound had an ID_50_ of 0.63 nmol/plant and thus showed higher inhibitory potential than the two phenylfuran derivatives. To understand binding of compound 14 in more detail, its structure was matched with that of BL. This showed that the O and N atoms of the isoxazole ring correspond to the two hydroxy groups on ring a of BL while the ester carbonyl oxygen corresponds to the carbonyl oxygen at C-6 in ring B of BL. The carbonyl oxygen of the second ester corresponds to the 22-OH group in the aliphatic side chain of BL. In addition, the benzene ring and the alkyl substituents of compound 14 correspond to hydrophobic features of BL [313]. Although the synthesis of compound 14 has not been described by Takimoto et al., it should be fairly simple since it is, in contrast to the phenylfuran derivatives shown in Figure 15, a symmetric compound.

#### 4.2.4. Turning an Antagonist to an Agonist: NSBR1

In an approach for converting a BL antagonist to a compound activating BRI1, Sugiura et al. [312] used the structures previously predicted by Takimoto et al. [313] as BL antagonists. In silico docking studies predicted that the previously identified compound 11 (Figure 17) binds well to BRI1, although it cannot interact with BAK1 due to lack of functional groups allowing formation of hydrogen bonds. 

Thus, the two proximate fluorine residues on the aromatic ring were replaced by hydroxy groups to obtain NSBR1 (Figure 17A). NSBR1 was synthesized by reaction of 3,4-difluorobenzoic acid with *N*,*O*-dimethylhydroxylamine hydrochloride in the presence of *N*,*N*′-dicyclohexylcarbodiimid to a Weinreb-Nahm amide, which was reacted with popylmagnesium bromide to the corresponding ketone. The 4-fluoro residue was substituted by piperazine and finally the obtained intermediate was reacted with 3,4-dihydroxybenzoic acid to the final product NSBR1 (Figure 17B). NSBR1 showed activity in the rice lamina inclination assay, confirming that it acts as a BL mimetic. However, its half maximal effective dose (ED_50_) with 0.8 nmol/plant was much higher than that of BL and CS, which showed ED_50_ values of 2.5 × 10^−5^ nmol/plant. In *A. thaliana*, it reduced the mRNA level of the BR biosynthesis genes *CPD* and *BR6OX2* if applied at concentration of 10–30 µM [312]. These data confirm that NSBR1 is a biologically active non-steroidal BL mimetic.

### 4.3. Compounds Targeting BIN2

#### 4.3.1. Lithium

Lithium, usually applied as lithium chloride, is long known for its effect on organ development in animals and slime molds. In the slime mold *Dictyostelium discoideum* lithium induces redifferentiation of prespore into prestalk cells [315]. In *Xenopus laevis* embryos, lithium triggers expansion of dorsal mesoderm, leading to duplication of the dorsal axis, which causes development of two-headed tadpoles [316]. In addition, lithium mimics insulin in mammals and stimulates glycogen synthase activity [317]. These effects could be addressed to inhibition of GSK-3β by lithium [318,319], which acts as a competitive inhibitor for binding of magnesium [320]. GSK-3β belongs to a conserved group of serine/threonine kinases present in all eukaryotes investigated so far. It is crucial for Wnt signaling, a regulatory pathway essential for cell fate determination and embryonic patterning in *Drosophila melanogaster* [321], axis formation in *Xenopus laevis* [322,323] and maintenance of skin and colon stem cell pluripotency in mammals [324]. In mammals, GSK-3β is also part of the insulin signaling pathway, where it regulates glycogen synthase activity [325]. While animals possess usually one or two GSK-3s, plants contain a gene family. For example, *Arabidopsis thaliana* possesses 10 and rice 9 GSK-3-like kinases with BIN2 being the best characterized member [126,127,128,129]. Lithium has been used in a number of studies for inhibiting BIN2 and its homologues in vitro [132,138,146] and in vivo [326,327,328,329], were it is usually applied in concentrations of 10 to 100 mM. However, lithium has substantial effects on plant growth. While very low concentrations may be beneficial for some plant species, elevated levels of lithium are toxic for plants [330] and induce necrosis along leaf margins, interveinal chlorosis and leaf abscission [331]. Lithium competes with plant nutrients including potassium, calcium and magnesium [332] and interferes with sodium/proton antiporter systems [333]. Lithium has also been demonstrated to inhibit inositol monophosphatase, which may lead to a reduction in free inositol levels. This may affect many plant functions including cell wall biosynthesis, auxin storage and transport and phosphorous storage [334]. The level of lithium tolerance varies significantly between plant species. For instance, avocado, soy bean and orange are highly sensitive and levels of less than 10 mg/kg of lithium in soil cause yield reduction by 25%. Other plant species like corn are more tolerant and a 25% yield reduction is observed at lithium levels of 70 mg/kg soil [331]. For *A. thaliana* grown in vitro, a presence of 15–20 mM lithium in the medium is lethal [335]. Because of these adverse effects, lithium is of limited use for inhibition of BIN2 and its homologues in plants, particularly for studies where a long-term application is required.

#### 4.3.2. Bikinin and Methyliodobikinin

Bikinin was identified in a screen of a 10,000 compound library for small molecules that cause constitutive BR responses in *A. thaliana* [70]. One compound designated bikinin (Figure 18A) induced responses including increased hypocotyl length, long petioles and light green leaves. Quantitative PCR revealed that BL as well as bikinin application repressed expression of *BR6OX2*, *CPD*, *DWF4* and other BR biosynthetic genes while the catabolic BR hydroxylase gene *BAS1* and the marker gene *SAUR-AC* were upregulated. The BR biosynthesis mutant *cpd* was rescued by both BL and bikinin while the BR signaling mutants *bri1-116* and *bin2-1* were only rescued by bikinin. This provided evidence that bikinin acts downstream of BL and, more specifically, at the level or downstream of BIN2. To investigate that in more detail, BIN2 and its homologues were cloned and expressed in *E. coli*. In vitro kinase assays using myelin basic protein as artificial substrate and [γ-^32^P]-ATP as co-substrate revealed that BIN2 and most of its homologues were inhibited by bikinin. In contrast, the MAP kinases AtMPK4 and AtMPK6 and 77 human protein kinases including GSK-3β were not inhibited by bikinin, showing that it is specific for BIN2 and its close homologues. Specificity of bikinin was confirmed by transcriptome analysis of BL and bikinin-treated plants, which showed pronounced overlap of the transcriptomes. In vitro kinase assays showed that bikinin-mediated inhibition of BIN2 can be diminished by increasing the ATP concentration, indicating that bikinin competes with ATP for binding to BIN2. Indeed, docking simulations showed that bikinin binds to the ATP pocket of BIN2. Interestingly, most amino acids in the ATP binding pocket predicted to interact with bikinin were conserved in the homologues of BIN2. A remarkable exception is tyrosine 117 (Y117), which is replaced by phenylalanine in ASKδ, a plant GSK-3β-like protein kinase not inhibited by bikinin. Exchanging Y117 of BIN2 to phenylalanine (mutant BIN2Y117F) lead to a bikinin-resistant version of BIN2, confirming that bikinin interacts with the ATP pocket of BIN2 and that Y117 is required for binding [70].

To investigate the structure-activity relationship of bikinin in more detail, a number of derivatives with altered substituents at the pyridine ring, a different position of the nitrogen in the pyridine ring and different length of the aliphatic side chain were synthesized [177]. Compounds with free carboxyl groups were obtained by refluxing the appropriate pyridlyamine in the presence of succinic anhydride using THF (tetrahydrofuran) as a solvent. Methylated compounds were obtained by reaction of the corresponding aminopyride with methylsuccinyl chloride in TFH in the presence of triethylamine. The preparation of bikinin and methyliodobikinin are illustrated in Figure 18A,B. In vitro kinase assays showed that replacing the bromo substituent by iodine increased the activity while altering the length of the aliphatic side chain, or the position of the nitrogen in the pyridine ring reduced it. Interestingly, esterifying the carboxyl group reduced the in vitro activity as assayed by in vitro kinase assays, but increased the in planta activity. That could be explained by different tissue permeabilities: uptake of the methylated form was more rapid and an approximately two-fold higher concentration in the plant tissue was reached. Rapid hydrolysis of the methylated form in plant tissues was observed, which resulted in the release of a more active free acid form. In addition, the in planta levels started to decline after a few hours, which was the accompanied by appearance of two additional peaks in the HPLC chromatogram, which could be identified as glutamic acid and malic acid conjugates of iodobikinin (Figure 19). SNG1, an enzyme of *A. thaliana* involved in modification of sinapic acid [336], was shown to be involved in conjugation of iodobikinin with malic acid [177]. Rapid modification of methyliodobikinin explains its transient effect for inducing BR responses in planta. Similar reactions are likely also responsible for inactivation of bikinin.

Based on the structures for bikinin and methyliodobikinin, three similar compounds designated A1, A2 and A19, where the halogen was replaced by nitrile or trifluoromethyl groups, were synthesized [337]. For biological activities no details were given except that treatment of French beans (*Phaseoulus vulgaris* cv. Fulvio) increased the length of the petiole of the second leaf by more than 10% and that treatment of wheat (*Triticum aestivum* cv. Arina) caused a curly root phenotype [337].

Since its introduction, bikinin has become the most popular compound in research for activating BR signaling downstream of BRI1. It is commercially available or can be prepared by a simple one step synthesis from readily available compounds with good yield [177]. It has proven to be very versatile and has been used in more than 100 studies. However, a disadvantage of bikinin and methyliodobikinin is their rapid loss of activity by metabolization in plant tissues [177].

#### 4.3.3. Brazide

In an attempt to replace the pyridine moiety of bikinin with other heterocycles, Liu and co-workers showed that a molecule named brazide that contained a 2-amino-5-methylthiazolyl moiety instead of the pyridyl residue showed substantially enhanced biological activity [166]. Brazide was prepared by reaction of 5-methyl-2-aminothiazole with ethylsuccinyl chloride in THF in the presence of triethylamine (Figure 20). Ethylsuccinyl chloride required for that reaction was prepared from succinic anhydride and chlorination of the intermediate by thionyl chloride.

*A. thaliana* hypocotyl elongation and rice lamina inclination assays indicated that brazide is more active than bikinin [166]. The activities of brazide and methyliodobikinin were not compared. Based on its similarity to bikinin it is expected that brazide inhibits BIN2 although that has not been verified experimentally so far. Recently, it was shown that pre-treatment of maize with brazide, bikinin or 24-epiBL protected the plant against toxicity from subsequently applied nicosulfuron, a herbicide frequently used in maize farming [338]. So far, it remains elusive whether brazide is metabolized in plant tissues in a similar way like methyliodobikinin.

### 4.4. Compounds with Unknown Targets

#### 4.4.1. KM-01

During a screen for BR-like substances from fungi, Kim et al. found that *Drechslera avenae*, the causal agent of leaf spot of oat (*Avena sativa*), caused severe inhibition of BL-induced lamina inclination in rice seedling explants [339]. Initial attempts for isolation of the bioactive compound gave a solid at a yield of 0.8 mg/L culture medium. The compound was named KM-01. A subsequent study showed that the yield can be significantly improved by growing the fungal cultures under blue light and by changing the extraction procedure from acidic to basic conditions. The modified method allowed isolation of 20.4 mg KM-01 per liter fungal culture. This amount allowed elucidation of the structure of KM-01 (Figure 21) by a combination of NMR, IR and circular dichroism spectroscopy, as well as chemical degradation and chiral HPLC [340]. In addition, it was found that KM-01 is also produced by *Pycnoporus coccineus*, a widely distributed saprophytic white rot fungus.

For evaluation of the biological activity of KM-01 the rice lamina inclination assay and hypocotyl elongation assays using radish (*Raphanus sativus* cv. Tokinashi) were used. Application of KM-01 on rice explants showed a decrease of the angle between leaf and sheath from 66° for the control to 45° in the presence on 10 µM KM-01, indicating that the activities of endogenous BRs were reduced. Rice seedling explants treated with 10 nM BL showed an angle of 146°, which was reduced to 59° by co-application of 0.1 µM KM-01 and to 36° by 10 µM KM-01. To assay the impact on hypocotyl elongation, 1 cm hypocotyl explants of radish seedlings were placed in test solutions containing BL and/or KM-01. BL (20 nM) showed a strong promoting effect on hypocotyl explant elongation, which was reduced by KM-01. Addition of KM-01 at a final concentration of 30 µM inhibited BL activity completely. Further bioassays indicated that KM-01 might be synergistic to GA_3_ while it might be antagonistic to ABA. No effect of KM-01 could be observed in bioassays for auxin, cytokinin for ethylene responses [339]. KM-01 is not commercially available and has not been applied since 1998, when its structure was elucidated [340]. Nevertheless, KM-01 might be an interesting compound for inhibiting BL activity.

#### 4.4.2. F1874-108

F1874-108 (Figure 21) and a number of similar compounds were identified in a screen for compounds that block ethylene response in *A. thaliana* [341]. Application of the compound in the range of 5–20 µM to light-grown *A. thaliana* seedlings increased the hypocotyl length significantly. Interestingly, co-application of the compound and BL showed strong synergism while F1874-108 alone had no impact on *bri1-6* mutants, which are defective in BR perception. In contrast, application of bikinin rescued *bri1-6* mutants. When BR production was limited by application of BRZ, plants co-treated with F1874-108 showed increased hypocotyl length. However, F1874-108-treated plants remained responsive to BRZ since increasing its concentration reduced hypocotyl length. Based on these results, the conclusion was drawn that F1874-108 increases the responsiveness to BL rather than it acts as a constitutive activator of BR signaling. This action is clearly different from the effect of any other compound interfering with BR signal transduction. BIN2 was suggested as target for F1874-108 [341] but no data confirming this assumption were reported. Thus, the mechanism how F1874-108 impacts on BR signaling remains elusive.

## 5. Conclusions

In the last two decades a number of compounds interfering with BR biosynthesis and signaling have been identified (Table 2). Some of them, BRZ and bikinin in particular, have been extensively used, for instance for the isolation of new mutants [69], investigation of crosstalk between BR signaling and other pathways [7,8,9,342], for studying the role of BRs in symbiosis between plants and fungi [343] as well as rhizobacteria [344], or for identification of BR-regulated chromatin remodeling factors [345,346]. A main reason for the popularity of inhibitors in research is their applicability to different genetic backgrounds and in many cases a wide range of plant species. For instance, the sterol biosynthesis inhibitor voriconazole has been shown to be active in more than 20 plant species from different classes including the monocots and many tribes of the dicots [78]. Similarly, BRZ has been applied for studying stress responses in tomato [347] and, fruit development and ripening in cucumber [348], tomato [349], strawberry [350] and grapes [351] as well as shooting control in fruit trees [352]. Bikinin and its derivatives have also been applied in a number of species including tomato [27], tobacco [353] and maize [338]. Performing such studies with mutants is more complicated or even impossible for species where no suitable mutants are available. Another important advantage of inhibitors is that they can overcome functional redundancy in polyploid species and gene families. In addition, inhibitor application can be limited to a specific tissue or developmental stage.

Ideally, inhibitors should be highly potent, specific and their target should be known. In addition, knowledge of pharmacological data like uptake kinetics and possible modification/inactivation reactions in the tissue is advantageous. Particularly the triazole-type BR biosynthesis inhibitors are highly potent with IC_50_ values below 1 µM, some of them even reach values in the range of 20 nM. Also, some compounds interfering with BL perception seem to be highly potent. Compounds interfering with proteins involved in BR signal transduction are generally less potent but bikinin and particularly methyliodobikinin have still IC_50_ values in the low µM range (Table 2).

For investigation of specificity, much effort has been done to dissect inhibition of BR and GA biosynthesis. Co-application of the inhibitor and BL (or 24-epiBL) as well was GA_3_ have proven to be a simple way to obtain indications about specificity for targeting BR and/or GA biosynthesis [167,169,170,214,215,217,220,278]. However, it is important to mention that a small response to GA_3_ is not necessarily a sign of unspecificity, since BR mutants, depending on the growth conditions, developmental stage and type of mutation, also show slight to pronounced GA responsiveness [190,191,192,354]. In contrast, less effort has been made to exclude inhibition of sterol biosynthesis.

This is surprising, since several lead compounds used for identification of BR biosynthesis inhibitors are well characterized inhibitors of fungal sterol biosynthesis. Moreover, several mutants with defects in sterol biosynthesis have phenotypes resembling BR mutants. For instance, *dwf1*, *dwf5* and *dwf7* are sterol biosynthesis mutants although, based on their phenotype, they are often referred to as BR biosynthetic mutants [40]. Likewise, compounds inhibiting BR biosynthesis, for instance BRZ, and sterol biosynthesis, for instance voriconazole, generate almost identical phenotypes in plants and both can be rescued by application of BL but not GA_3_ in seedlings [8]. Possible inhibitory activities on sterol biosynthesis should be investigated more thoroughly in the future. Similarly, ABA catabolism might be an off-target of BR biosynthesis inhibitors. The triazole uniconazole has been identified as an efficient 8′-ABA-hydroxylase inhibitor while BRZ had only minute inhibitory effects on that enzyme [199]. For the other BR biosynthesis inhibitors no information about their effect on 8′-ABA-hydroxylase is available.

For a number of BR biosynthesis inhibitors the targets were identified, which were DWF4, CYP90D1 or DET2. In addition, voriconazole causes similar phenotypes but targets CYP51, a cytochrome P450 oxidase involved in sterol biosynthesis (Table 2). Some of results regarding the target sites are surprising, mainly that (*2S*,*4R*)-Brz220 was shown to bind to DWF4 while YCZ-18 was reported to interact with CYP90D1, although both compounds have a similar structure and possess a dioxolane ring. However, it must be emphasized that binding of CYP90D1 by (*2S*,*4R*)-Brz220 and of DWF4 by YCZ-18 has not been experimentally tested. Thus, it cannot be excluded that (*2S*,*4R*)-Brz220 and YCZ-18 interact with both, DWF4 and CYP90D1.

A number of steroidal and non-steroidal compounds have been shown to inhibit DET2 in *A. thaliana* and tomato. However, compared to triazole inhibitors, they are poorly characterized. Detailed investigation of their potency at the phenotypic and molecular level and identification of more potent inhibitors of plant steroid 5α-reductases are challenges for the future. Also identification of compounds targeting further enzymes of BR biosynthesis would clearly be useful tools for research. While other triazoles might be candidates for inhibitors of the cytochrome P450s CPD, BR6OX1 and BR6OX2, spironolactone [285] and brassinopride [298] might potentially inhibit other types of enzymes participating in BR biosynthesis.

Moreover, for BR signaling only a few targets are known: lithium, bikinin and its derivatives methyliodobikinin and (presumably) brazide target BIN2 and its homologues while brassinolide-2,3-acetonide and a number of non-steroidal compounds target the BR receptor complex BRI1/BAK1. Among them, only bikinin and lithium have been used frequently, although the recently discovered BRI1 inhibitors and activators might be valuable tools for future research. Lithium is toxic for plants and thus of limited value. Bikinin seems to be quite specific since it did not target other protein kinases and the transcriptome of plants treated with BL and bikinin showed extensive overlap [70]. However, a drawback of bikinin and its derivative methyliodobikinin is their rapid inactivation in plant tissue by conjugation with glutamic acid and malic acid [177]. Similar inactivation mechanisms might be expected for the other bikinin derivatives including brazide since they share the succinic acid side chain where conjugation occurs. A main challenge for the future will be the identification of bikinin derivatives with increased half-life in plant tissues.

Compounds inhibiting enzymes involved in BR biosynthesis or BR signaling have been shown to be valuable tools in research. Nowadays, more crop plants become the focus of research. This will increase the importance of such compounds since they are potentially active in many plant species. Thus, small molecules interfering with BR biosynthesis and signaling will continue to play a major role in BR research in the future.

## Figures and Tables

**Figure 1 molecules-24-04372-f001:**
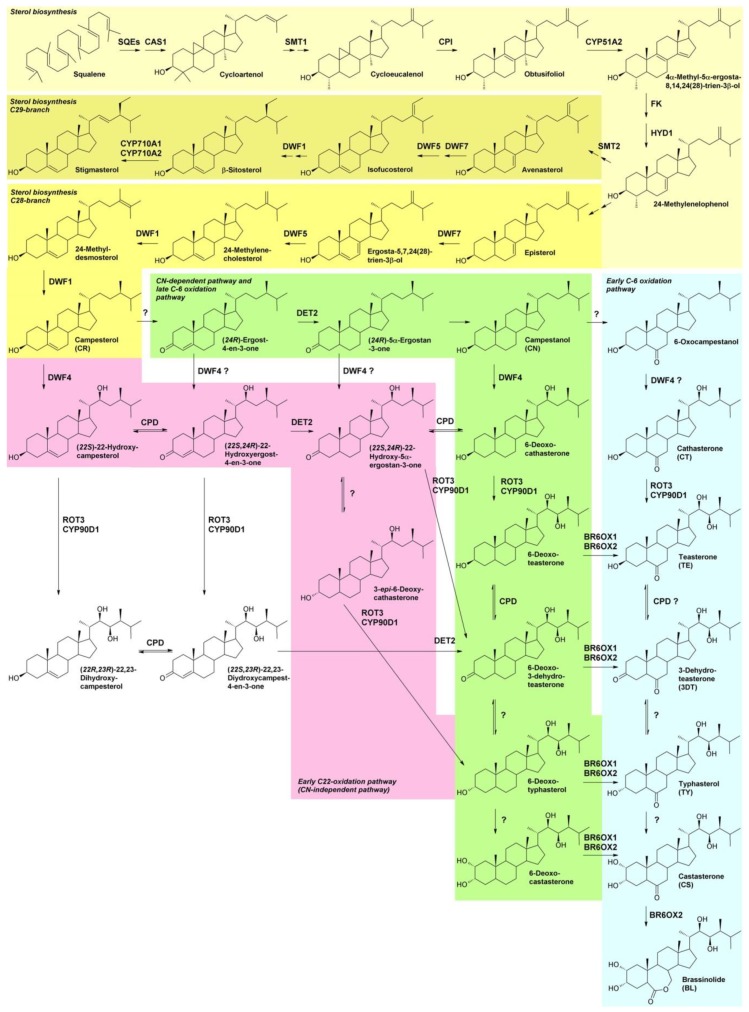
Simplified scheme of sterol and BR biosynthesis. In sterol biosynthesis, squalene is converted in a series of reactions to 24-methylenelophenol, where the pathway splits into the C29-branch leading to β-sitosterol and stigmasterol as end products and into the C28-branch with campesterol, the precursor of BR biosynthesis, as a final product. In *A. thaliana*, the early C-22 oxidation pathway seems to be the predominant route of BR biosynthesis, which is, however, interconnected with late C-6 oxidation at multiple steps. The sterol synthesis pathway was adapted from [40] and the BR biosynthesis pathway from [59,62,63].

**Figure 2 molecules-24-04372-f002:**
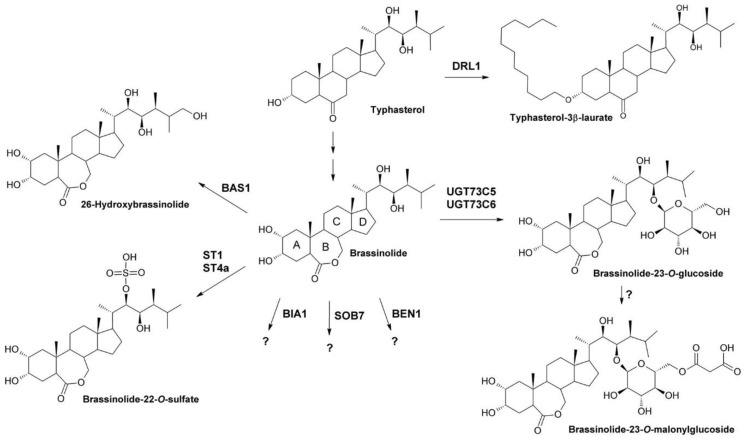
Mechanisms of BR inactivation. The acyltransferase DRL can acylate the BL precursor typhasterol with lauric acid and probably also myristic and palmitic acid residues. The modified hydroxy group has not been analyzed in *A. thaliana* but data from *Ornithopus sativus* suggest that O-3 might be modified. BL (and CS) can be glucosylated by the UDP-glucosyltransferases UGT73C5 and UGT73C6. The product, brassinolide-23-*O*-glucoside, can be malonylated by a yet unknown enzyme. The sulfotransferases ST1 and ST4a can modify BL (and CS) at O-22 although they are significantly more active on 28-homobrassinolide. The cytochrome P450 BAS1 can hydroxylate BL (and CS) at C-26. The reactions catalyzed by BIA1, SOB7 and BEN1 remain so far unknown. Numbering of the rings is included in the structure of BL.

**Figure 3 molecules-24-04372-f003:**
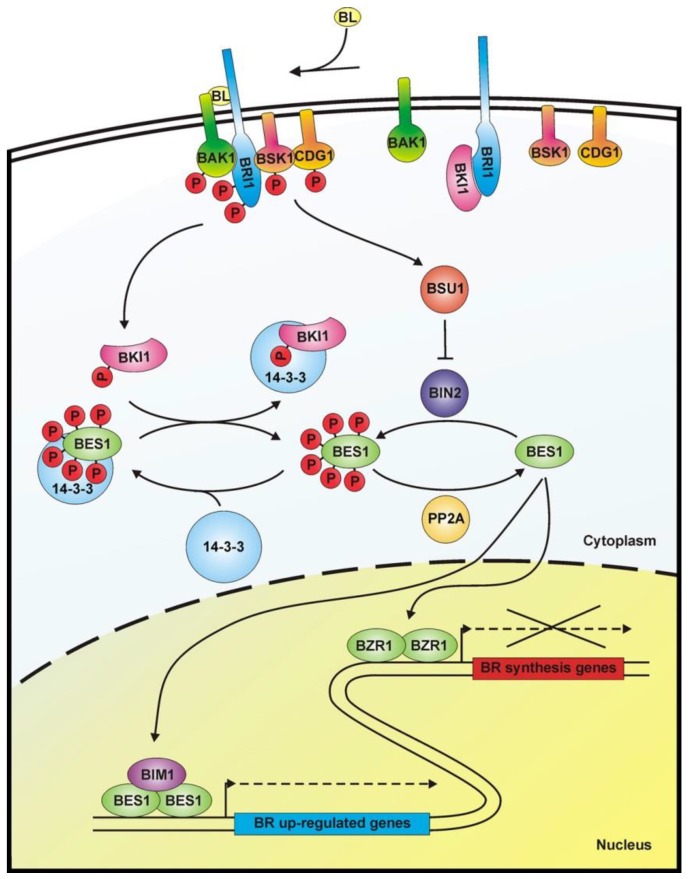
Simplified scheme of BR signal transduction. Binding of BL to BRI1/BAK1 triggers phosphorylation of the receptor complex and dissociation of BKI1. Subsequently, BSK1/CDG1 are phosphorylated by activated BRI1 and the signal is transduced to the protein phosphatase BSU1, which dephosphorylates and thereby inactivates BIN2. Now BIN2 can no longer phosphorylate BES1 and its homologues, which accumulate in their dephosphorylated form in the nucleus and regulate gene expression, either on their own or in complex with bHLH transcription factors, for instance BIM1. Phosphorylated BES1 is bound by 14-3-3 proteins and thereby kept in the cytoplasm. Free BKI1 can sequestrate 14-3-3 proteins and thereby release BES1/BZR1-type transcription factors.

**Figure 4 molecules-24-04372-f004:**
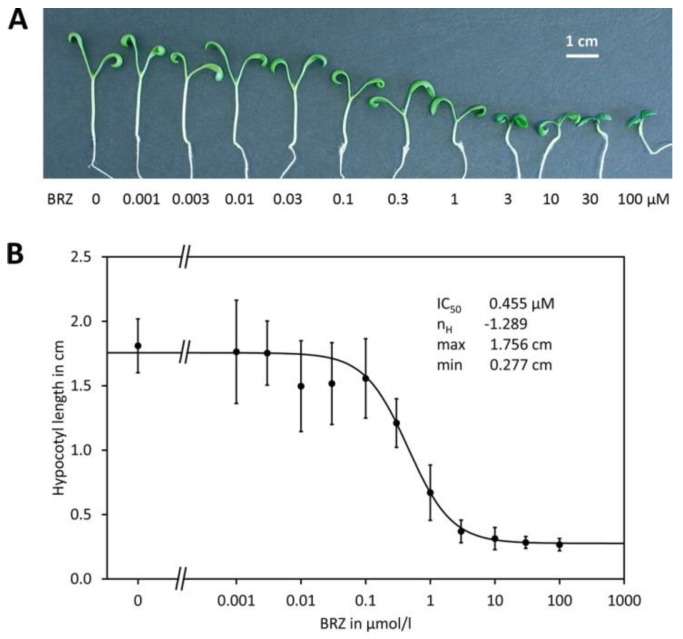
IC_50_ determination of BR biosynthesis inhibitors. (**A**) Picture of representative cress plants grown on ½ MS medium containing 1% sucrose and supplemented with the indicated concentrations of brassinazole (BRZ). The plants were grown under long day conditions (16 h light at 80 µmol∙s^−1^∙m^−2^ and 8 h dark) at 21 °C for 7 d. (**B**) Calculation of the IC_50_ using logit regression. Each point and error bar represents the length and SD of 30 seedlings. The original data and the calculation are presented in Appendix A.

**Figure 5 molecules-24-04372-f005:**
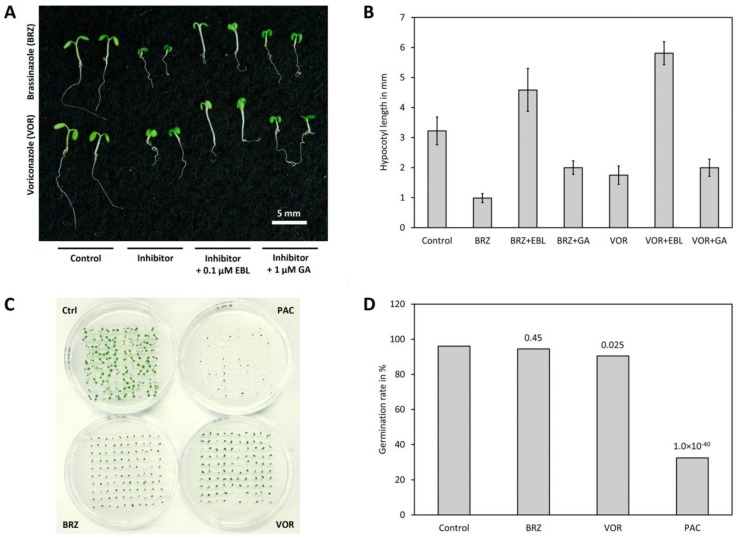
Phenotype-based methods for assaying specificity of BR biosynthesis inhibitors. (**A**) Co-application of the BR biosynthesis inhibitor BRZ and the sterol biosynthesis inhibitor voriconazole with 0.1 µM 24-epiBL (EBL) and 1 µM GA_3_ (GA) for investigating specificity. Overall phenotype and (**B**) Quantification of the hypocotyl length. (**C**) Assaying germination rate on ½ MS medium containing 1% sucrose (Ctrl) and in the presence of 10 µM paclobutrazol (PAC), a GA biosynthesis inhibitor, as well as 10 µM BRZ and 10 µM voriconazole (VOR). Overall phenotype and (**D**) Quantification of the germination rate calculated from at least 200 seeds. The numbers above the column represent the χ^2^-test *p*-values for germination rates compared to the control. Original data are included in Appendix A.

**Figure 6 molecules-24-04372-f006:**
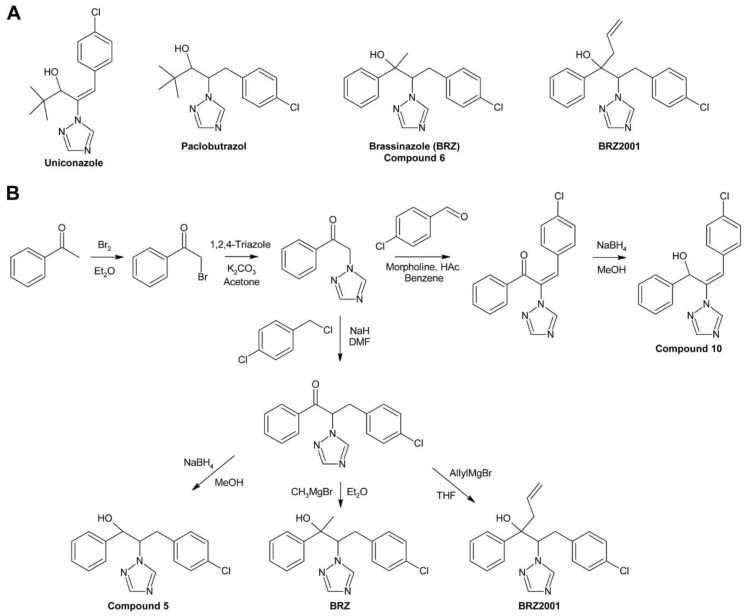
BRZ and BRZ2001. (**A**) Structures of uniconazole, paclobutrazol, brassinazole (BRZ) and BRZ2001. (**B**) Synthesis of BRZ, BRZ2001 and compounds 5 and 10. Abbreviations: DMF, *N*,*N*-dimethylformamide; Et_2_O, diethylether; HAc, acetic acid; MeOH, methanol; THF, tetrahydrofurane. Numbering of compounds 5 and 10 refers to [214].

**Figure 7 molecules-24-04372-f007:**
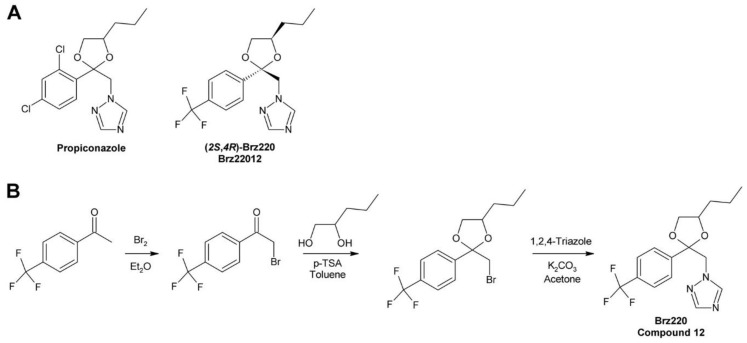
Brz220. (**A**) Structures of propiconazole and (*2S*,*4R*)-Brz220, most potent stereoisomer of Brz220. (**B**) Synthesis of Brz220. Abbreviations: Et_2_O, diethylether; pTSA, p-toluenesulfonic acid. Numbering of compound 12 refers to [167].

**Figure 8 molecules-24-04372-f008:**
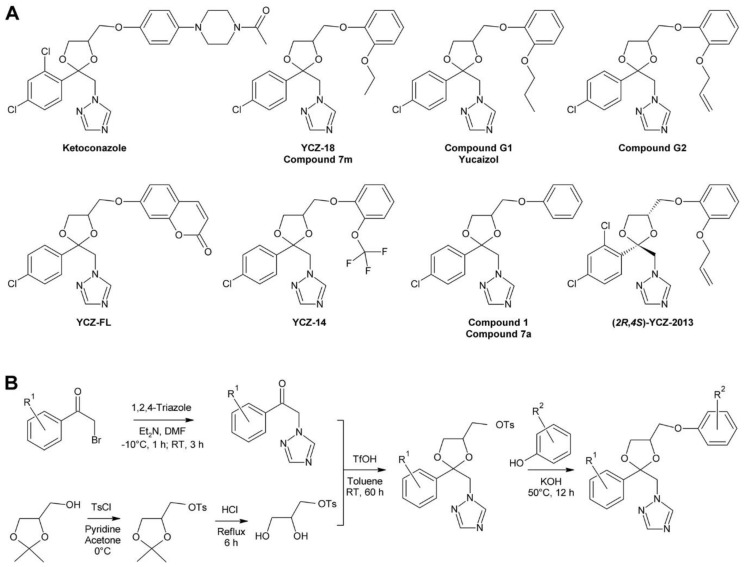
The BR biosynthesis inhibitors of the YCZ series. (**A**) Structures of ketoconazole and representatives of the YCZ series. (**B**) General scheme for synthesis of BR biosynthesis inhibitors of the YCZ type. Abbreviations: Et_3_N, triethylamine; DMF, *N*,*N*-dimethylforamide; RT, room temperature; TfOH, trifluoromethanesulfonic acid; TsCl, tosyl chloride. Numbering of compound 7m refers to [170], of G1 and G2 to [169] and of compound 1/7a to [170,219].

**Figure 9 molecules-24-04372-f009:**
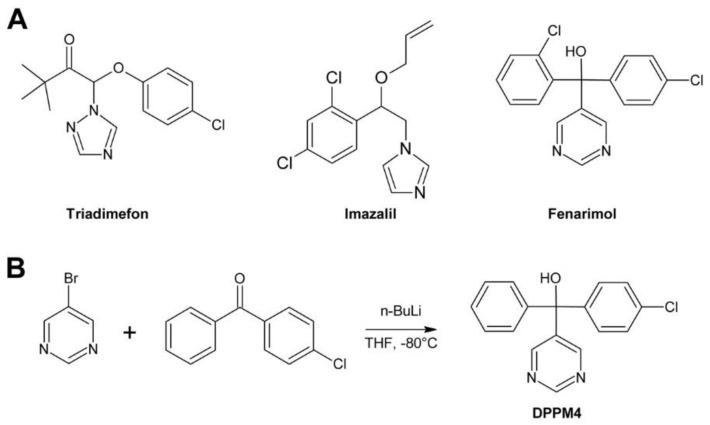
Further heterocyclic BR biosynthesis inhibitors. (**A**) Structures of the triazole triadimefon, the imidazole imazalil, and the pyrimidine fenarimol. (**B**) Synthesis of DPPM4. Abbreviation: THF, tetrahydrofuran.

**Figure 10 molecules-24-04372-f010:**
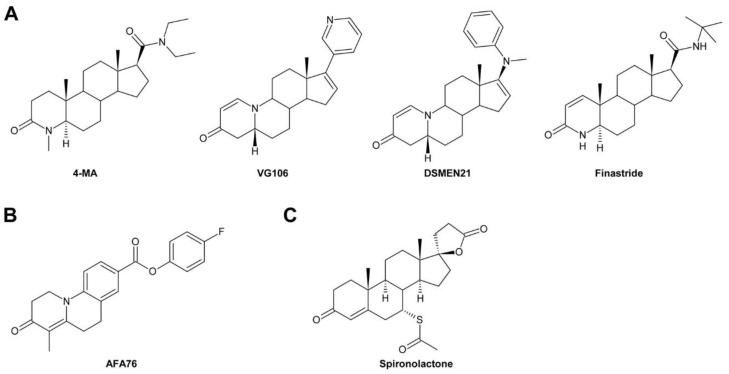
Steroidal and non-steroidal BR biosynthesis inhibitors. (**A**) Structures of the steroidal 5α-reducatse inhibitors 4-MA, VG106, DSMEN21 and Finastride. (**B**) Structure of the non-steroidal 5α-reducatse inhibitor AFA76. (**C**) Structure of spironolactone, a steroidal BR biosynthesis inhibitor with unknown target.

**Figure 11 molecules-24-04372-f011:**
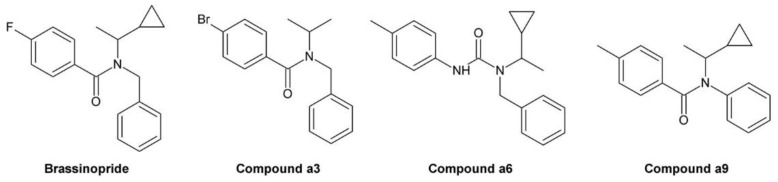
Brassinopride and related compounds. Brassinopride and compound a3 are most active for inhibition of hypocotyl elongation of dark grown *A. thaliana* while compound a6 and a9 are less active. All four compounds promote apical hook formation in dark grown *A. thaliana* seedlings. Numbering of compounds a3, a6 and a refers to [298].

**Figure 12 molecules-24-04372-f012:**
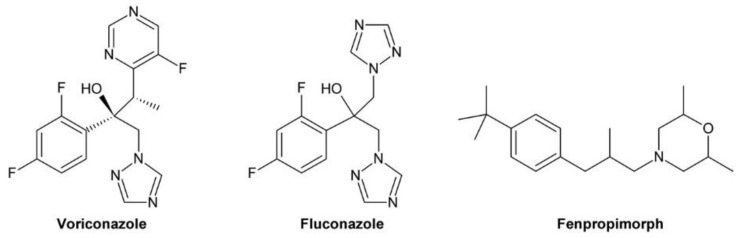
Compounds inhibiting sterol biosynthesis in plants. Structures of the triazole CYP51 inhibitors voriconazole, fluconazole and of the cycloeucalenol-obtusifoliol isomerase inhibitor fenpropimorph.

**Figure 13 molecules-24-04372-f013:**
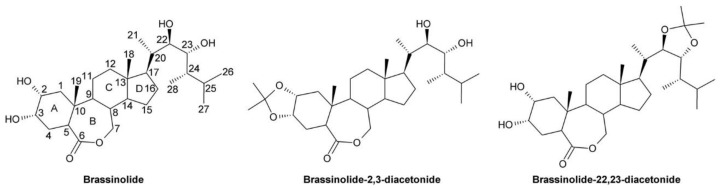
Structures of the brassinolide, brassinolide-2,3-acetonide and brassinolide-22,23-acetonide. Numbering of the rings and carbon atoms is included in the structure for brassinolide.

**Figure 14 molecules-24-04372-f014:**
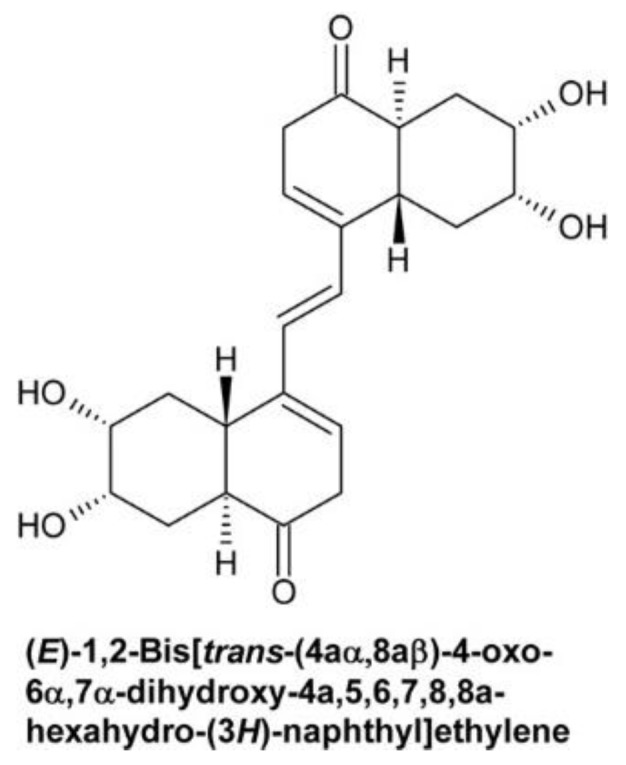
Structure of (*E*)-1,2-bis[*trans*-(4aα,8aβ)-4-oxo-6α,7α-dihydroxy-4a,5,6,7,8,8a-hexahydro- (3*H*)-naphthyl]-ethylene.

**Figure 15 molecules-24-04372-f015:**
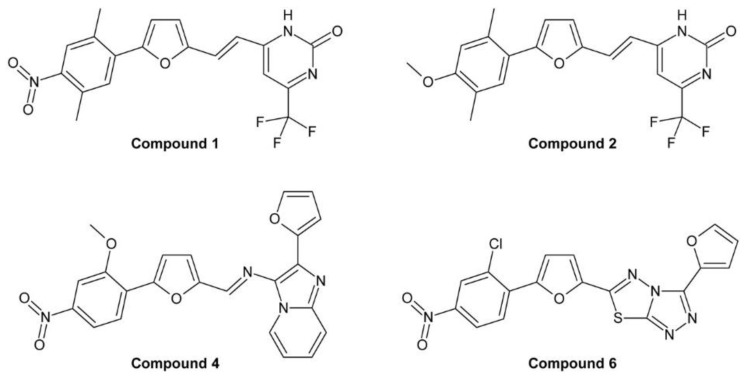
Structure of phenylfuran derivatives impacting on BL perception. Compound 1 and 2 act as BL antagonists suppress BL perception while compound 4 and 6 were reported to act as BL mimetics activating BR signaling. Numbering of compounds 1 and to refers to [313] and numbering of compounds 4 and 6 to [314].

**Figure 16 molecules-24-04372-f016:**
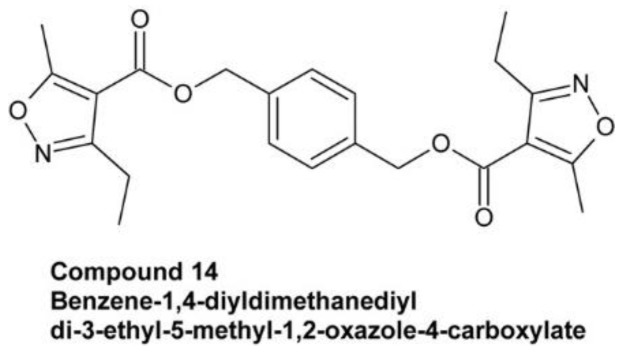
Structure of benzene-1,4-diyldimethanediyl di-3-ethyl-5-methyl-1,2-oxazole-4- carboxylate. Numbering of compound 14 refers to [313].

**Figure 17 molecules-24-04372-f017:**
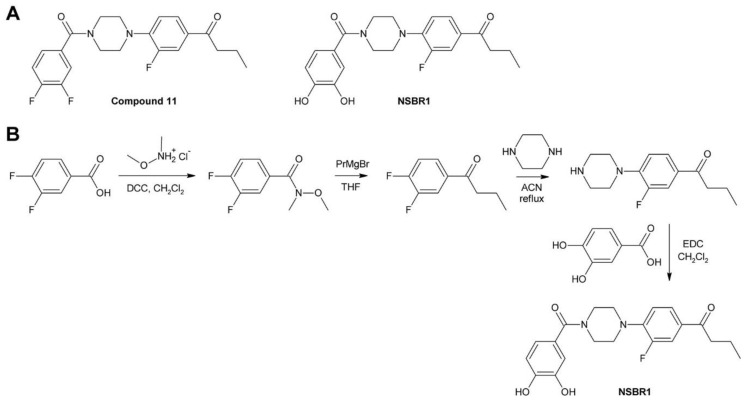
NSBR1. (**A**) Structures of compound 11 and NSBR1, which was designed in silico from compound 11 by replacing the two proximate fluorine atoms by hydoxy groups. (**B**) Synthesis of NSBR1. *Abbreviations*: ACN, acetonitrile; DCC, *N*,*N*′-dicyclohexylcarbodiimide; EDC, 1-ethyl-3-(3- dimethylaminopropyl)carbodiimide; PrMgBr, propylmagnesium bromide. Numbering of compound 11 refers to [313].

**Figure 18 molecules-24-04372-f018:**
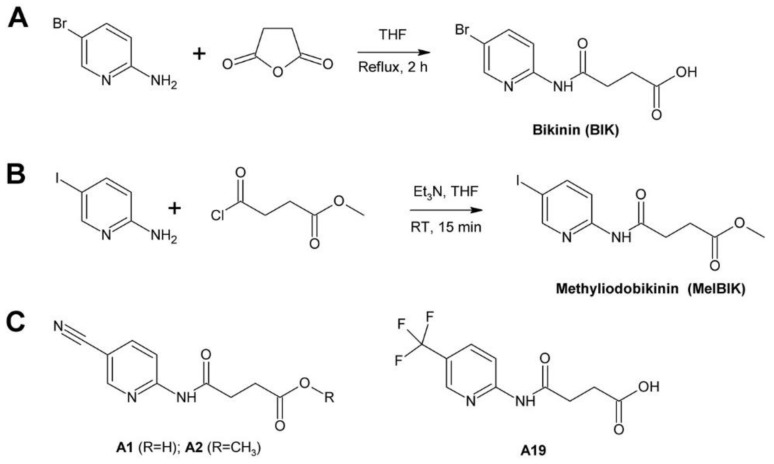
Bikinin and its derivatives. (**A**) Synthesis of bikinin. (**B**) Synthesis of methyliodobikinin. Abbreviations: Et_3_N, triethylamine; RT, room temperature; THF, tetrahydrofuran. (**C**) Structures of further bikinin derivatives with biological activity.

**Figure 19 molecules-24-04372-f019:**
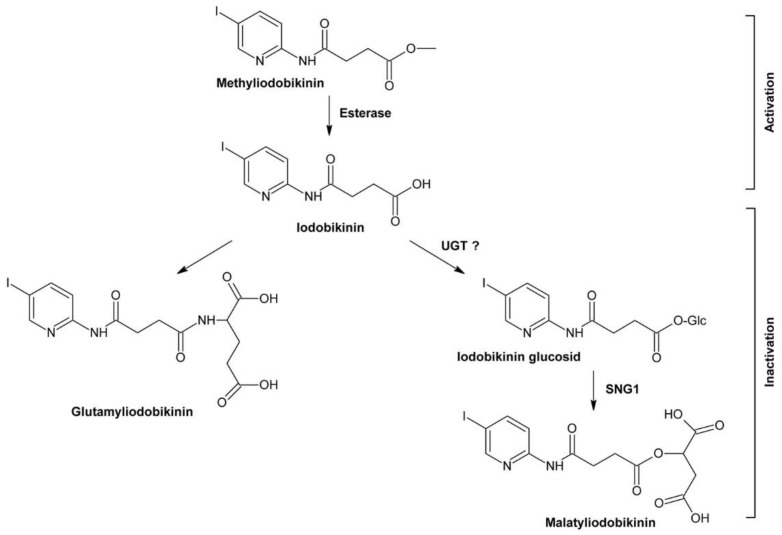
Metabolization of methyliodobikinin. The hydrophobic compound methyliodobikinin enters the cell and is rapidly hydrolyzed, presumably by the action of esterases, to the active form iodobikinin, which can inhibit BIN2 and most of its homologues. Iodobikinin is inactivated by conjugation with glutamic acid or malic acid. For the latter, iodobikinin glucoside may act as an intermediate although direct evidence for the presence of that compound in methyliodobikinin- treated plants is missing so far. The figure was adapted from [177].

**Figure 20 molecules-24-04372-f020:**
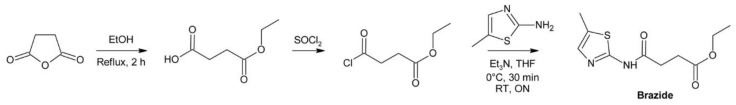
Synthesis of brazide. *Abbreviations*: EtOH, ethanol; Et_3_N, triethylamine; ON, overnight; RT, room temperature; THF, tetrahydrofuran.

**Figure 21 molecules-24-04372-f021:**
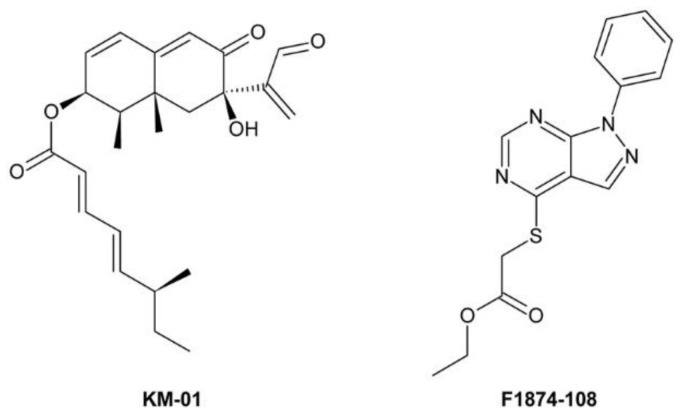
Structures of KM-01 and F1874-108.

**Table 1 molecules-24-04372-t001:** Enzymes involved in sterol and BR biosynthetic pathways in *A. thaliana*.

Abbreviation; Alternative Name	Name	*A. thaliana* Locus	References
BR6OX1; CYP85A1	BRASSINOSTEROID-6-OXIDASE 1	AT5G38970	[30]
BR6OX2; CYP85A2	BRASSINOSTEROID-6-OXIDASE 2	AT3G30180	[30,46]
CPD; CYP90A1	CONSTITUTIVE PHOTOMORPHOGENIC DWARF	AT5G05690	[15]
CAS1	CYCLOARTENOL SYNTHASE 1	AT2G07050	[47]
CPI1	CYCLOPROPYL ISOMERASE	AT5G50375	[48]
CYP51A2; CYP51G1	CYTOCHROME P450 51A2	AT1G11680	[42]
CYP90D1	CYTOCHROME P450 90D1	AT3G13730	[49]
CYP710A1	CYTOCHROME P450 710A1	AT2G34500	[50]
CYP710A2	CYTOCHROME P450 710A2	AT2G34490	[50]
DET2; DWF6	DEETIOLATED 2	AT2G38050	[18,51]
DWF1; DIM; CBB1	DWARF 1	AT3G19820	[35,36]
DWF4; CYP90B1	DWARF 4	AT3G50660	[52]
DWF5	DWARF 5	AT1G50430	[37]
DWF7; BUL1; STE1	DWARF 7	AT3G02580	[38]
FK; HYD2	FACKEL	AT3G52940	[43,44]
HYD1	HYDRA 1	AT1G20050	[45]
ROT3; CYP90C1	ROTUNDIFOLIA 3	AT4G36380	[49,53]
SMT1	STEROL METHYLTRANSFERASE 1	AT5G13710	[41]
SMT2	STEROL METHYLTRANSFERASE 2	AT1G20330	[39]
SQE1	SQUALENE EPOXIDASE1	AT1G58440	[54]
SQE2	SQUALENE EPOXIDASE2	AT2G22830	[54]
SQE3	SQUALENE EPOXIDASE3	AT4G37760	[54]

**Table 2 molecules-24-04372-t002:** Overview of small molecules interfering with BR biosynthesis, perception or signaling.

Compound	Type	Commercially Available	IC_50_ in µM ^1^	Target; *K_d_* in µM ^2^
BRZ	Triazole	Yes	0.58 ± 0.38 At [170]; 0.73 ± 0.13 At L 5d [235]; 0.77 At D 6 d [217]	DWF4; 1.05 [174]
BRZ2001	Triazole	No	0.55 At [219]; 0.33 At D 6 d [217]	DWF4 [221]
BRZ220	Triazole	No	0.61 ± 0.14 At [169]	DWF4 [168]
(*2S*,*4R*)-BRZ220	Triazole	No	0.01 C L 8 d [224]; 1.21 At D 5d [168]	DWF4; 0.22 [168]
YCZ-18	Triazole	No	0.10 ± 0.03 At [170]; 0.13 ± 0.03 At [169]	CYP90D1; 0.79 [73]
YCZ-14	Triazole	No	0.12 ± 0.04 At L 5 d [235]	CYP90D1 (?)
(*2S*,*4R*)-YCZ-2013	Triazole	No	0.024 ± 0.003 ^3^ At [236]	CYP90D1 (?)
YCZ-FL	Triazole	No	1.56 At D 5 d [234]	CYP90D1 (?)
Yucaizol	Triazole	No	0.8 R L 10 d [163]; 0.045 ± 0.003 At D 5 d [169]	CYP90D1 (?)
Propiconazole	Triazole	Yes	-	CYP90D1; 0.76 [75]
Triadimefon	Triazole	Yes	Approximately 3 ^4^ [175]	DWF4; 2.5 [175]
Imazalil	Imidazole	Yes	3.44 At D 6 d [217]	-
Fenarimol	Pyrimidine	Yes	1.8 ± 02 At D 5 d [74]	CYP90D1; 0.79 [74]
DPP4	Pyrimidine	No	-	CYP90D1 (?)
4-MA	Steroid	No	-	DET2; *K_i_* 0.3 [76,77]
Finasteride	Steroid	Yes	-	DET2 [76]
AFA76	Benzo[c]quinolizin-3-one	No	-	DET2 [76]
Spironolactone	Steroid	Yes	Approximately 30 ^5^ [285]	-
Brassinopride	Amide	No	17 At D [298]	-
Voriconazole	Triazole	Yes	0.083 ± 0.019 C L 7 d ^6^	CYP51 [78]
Lithium chloride	Salt	Yes	-	BIN2 [132,138]
Brassinolide-2,3-acetonide	Steroid	No	-	BRI1 [310]
Compounds 1, 2, 4 and 6	Phenylfuran	No	Comp. 1, ID_50_: 5 nmol/plant RLI; Comp. 2, ID_50_: 3.2 nmol/plant RLI [313]	BRI1 [313,314]
Compound 14 ^7^	-	No	ID_50_: 0.63 nmol/plant RLI [313]	BRI1 [313]
NSBR1	-	No	ED_50_: 0.8 nmol/plant RLI [312]	BRI1 [312]
Bikinin	Pyridine	Yes	EC_50_: 23.3 At L 6 d [177]	BIN2; 241 [70]
Methyliodobikinin	Pyridine	No	EC_50_: 6.8 At L 6 d [177]	BIN2 [177]
Brazide	Thiazole	No	-	BIN2 (?)
KM-01	Natural compound	No	-	-
F1874-108	-	No	-	BIN2 (?) [341]

^1^*Abbreviations*: IC_50_, half maximal inhibitory concentration; ID_50_, half maximal inhibitory dose; EC_50_, half maximal effective concentration; ED_50_, half maximal effective dose; At, *A. thaliana* hypocotyl elongation assay; C, cress hypocotyl elongation assay; R, rice seedling retardation assay; RLI, rice lamina inclination assay; D, grown in the dark; L, grown in long day conditions; d, grown for the indicated number of days. BIN2 means BIN2 and its homologues. ^2^ A question mark (?) indicates that the target is expected but experimental evidence is lacking. Abbreviations: *K_d_*, dissociation constant; *K_i_*, inhibition constant. ^3^ The (*2S*,*4R*) enantiomer has approximately the same activity. ^4^ Estimated from Figure 5 of the cited reference. ^5^ Estimated from Figure 3 of the cited reference. ^6^ This study (see Appendix A).^7^ Benzene-1,4-diyldimethanediyl di-3-ethyl-5-methyl-1,2-oxazole-4-carboxylate.

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
