# Peer review of "Inhibitors of Brassinosteroid Biosynthesis and Signal Transduction"

_molecules, 2019, doi:10.3390/molecules24234372_

Round 1

Reviewer 1 Report

In this manuscript, after summarizing the current knowledge of BR biosynthesis, metabolism, and receptor signaling, the method for evaluating the activity of BR regulators is explained in detail, and then each regulator is described carefully. Overall, it is very well written and has great value to be used as a bible for BR control agents. Here are some points I noticed about the content.

P17, L613-675.

In this paragraph, the authors state that there are some unclear points about the relationship between the absolute configuration of BRZ and the activity. As described by the authors, the chemical structure of BRZ has been depicted as 2S3R in many papers. I believe that this origin is in Fig. 1 of the paper by Asami et al (ref 207 in the manuscript). Asami et al presented the 2S3R structure as BRZ, but the racemic mixture was actually used. The major ingredient of all the reagents marketed as BRZ is the racemic form of BRZ. Commercially available BRZ is more than 93-98% in purity by HPLC, so it seems that diastereomers are hardly contained. Therefore, all research results using commercially available BRZ should be considered to have been obtained with racemic BRZ (SR/RS mixture) with little SS/RR. I think that 2R3S, even 2S3R, has never been tested for its activity. Only the racemate will be evaluated for activity.

In the crystal structure of CYP90B1-BRZ, the binding molecule has the (2R,3S) configuration. The BRZ used in the co-crystal will also be commercially available BRZ (although the source is not specified). As a result, the co-crystal were obtained with (2R,3S) instead of (2S,3R), so the active configuration of BRZ is likely to be (2R,3S), as described by the authors.

Based on the above, in the future, I think that it will be necessary to specify that it is an SR/RS racemic mixture when researchers publish the results using commercial BRZ. In particular, when drawing the chemical structure of BRZ, it is important to draw using a wedge-shaped bond in the sense of specifying the relative configuration. But at the same time, it should be specified that it is a relative configuration, not an absolute configuration. I think it would be nice if the authors could write these things in this manuscript.

Author Response

Reviewer 1

In this manuscript, after summarizing the current knowledge of BR biosynthesis, metabolism, and receptor signaling, the method for evaluating the activity of BR regulators is explained in detail, and then each regulator is described carefully. Overall, it is very well written and has great value to be used as a bible for BR control agents. Here are some points I noticed about the content.

P17, L613-675.

In this paragraph, the authors state that there are some unclear points about the relationship between the absolute configuration of BRZ and the activity. As described by the authors, the chemical structure of BRZ has been depicted as 2S3R in many papers. I believe that this origin is in Fig. 1 of the paper by Asami et al (ref 207 in the manuscript). Asami et al presented the 2S3R structure as BRZ, but the racemic mixture was actually used. The major ingredient of all the reagents marketed as BRZ is the racemic form of BRZ. Commercially available BRZ is more than 93-98% in purity by HPLC, so it seems that diastereomers are hardly contained. Therefore, all research results using commercially available BRZ should be considered to have been obtained with racemic BRZ (SR/RS mixture) with little SS/RR. I think that 2R3S, even 2S3R, has never been tested for its activity. Only the racemate will be evaluated for activity.

In the crystal structure of CYP90B1-BRZ, the binding molecule has the (2R,3S) configuration. The BRZ used in the co-crystal will also be commercially available BRZ (although the source is not specified). As a result, the co-crystal were obtained with (2R,3S) instead of (2S,3R), so the active configuration of BRZ is likely to be (2R,3S), as described by the authors.

Based on the above, in the future, I think that it will be necessary to specify that it is an SR/RS racemic mixture when researchers publish the results using commercial BRZ. In particular, when drawing the chemical structure of BRZ, it is important to draw using a wedge-shaped bond in the sense of specifying the relative configuration. But at the same time, it should be specified that it is a relative configuration, not an absolute configuration. I think it would be nice if the authors could write these things in this manuscript.

Response:

Many thanks for the positive evaluation of our manuscript and for the helpful comments. In the revised manuscript we have mentioned that in the cited studies the racemic mixture was used despite that the structure of (2S,3R)-BRZ was shown. In addition, we have mentioned that reporting the stereochemistry of the applied BRZ is advisable.

The comments about purity of commercially available BRZ are also very interesting. However, we could not find any citable reference for that and thus we decided not mentioning that the in the manuscript.

Changes in the manuscript were marked in yellow.

Reviewer 2 Report

Inhibition of BR biosynthesis is an interesting field of research that has attracted a lot of effort during the last decades. In this context this review of small molecules exhibiting this activity will be useful for many readers of Molecules. This review is comprehensive and well organized, and the presented data is discussed properly. Thus, I recommend publishing this work after all points detailed below, and the attached file, have been addressed.

Line 62, 63, 62, 65 Each phrase in these lines requires references

Line 134, 137. Phrase needs reference

Line 228. For recognition of BL by its receptor I think it is important to include the following reference. She, J., Z. Han, B. Zhou and J. Chai (2013). "Structural basis for differential recognition of brassinolide by its receptors." Protein & Cell 4(6): 475-482.

Line 275. This paragraph needs some references

Line 370. I think that the following reference should be added to 153. Li, H., H. Wang and S. Jang (2017). "Rice Lamina Joint Inclination Assay." Bio-protocol 7: e2409.

Line 385. In this paragraph one reference is needed at least.

Line 440. The phrase “Using such assays, the inhibitory effects of a number of steroidal and non-steroidal compounds on DET2 [169,170].” must be changed to “Using such assays, the inhibitory effects of a number of steroidal and non-steroidal compounds on DET2 [169,170] were determined”

Line 492. It reads Michaelis-Mention. Should be corrected, Same line “the” must de deleted

Line 504. It should read “similar phenotype characterized by reduced hypocotyl…”

Line 522. Results shown in Figure 5A and 5B are from literature or were measure in this work?? Specify in the text.

Line 535. Same as previous point

Line 564. Statements made in paragraph starting in this line require references

Line 566. “biosynthesis and responses and for manipulating BR levels…”. Delete

Line 585. Figure 6. Triazol derivatives must be numbered in correlative form. Compounds 5 and 10 are the only ones with numbers. These numbers must be used in the text

Line 649. The phrase “whose effected proteins act transcription factors in…..” should be rewritten

Line 658. Figure 7. Compounds must be numbered properly.

Line 662. “the reaction sequence for the example of Brz220.”  should be change to “the reaction sequence for the synthesis of Brz220.”

Line 665. Number of compound 12 should adapted to right numeration.

Line 689. It reads “Another series of BR biosynthesis inhibitors possessing a dioxolane ring….”. I think it is necessary to stress that this series are also triazol derivatives. For example, “Another series of BR biosynthesis inhibitors possessing a dioxolane ring, in addition to the triazol ring,…”

Line 694. Figure 8. Same as above, compounds should be numbered following the sequence as they appear in the text.

Line 754. The phrase “This allowed using published data from synthesis of kentoconazole as a reference, which facilitated determination of the absolute configuration” should be rewritten

Line 962. Figure 11. Same as above, compounds should be numbered following the sequence as they appear in the text.

Line 1001. “another means” should be changed “another way”

Line 1017. Add the following reference “Liu, J., D. Zhang, X. Sun, T. Ding, B. Lei and C. Zhang (2017). "Structure-activity relationship of brassinosteroids and their agricultural practical usages." Steroids 124: 1-17.”

Line 1080. Numeration of compounds must be corrected.

Line 1086. Figure 15. Same as above, compounds should be numbered following the sequence as they appear in the text

Line 1102. Figure 16. Same as above, compounds should be numbered following the sequence as they appear in the text

Line 112. Figure 17. Same as above, compounds should be numbered following the sequence as they appear in the text

Line 1118. Change to “Compound NSBR1 was synthesized….”

Line 1215. It reads “three similar compounds designated 1215 A1, A2 and A19, where the halogen was replaced by nitrile or trifluoromethyl groups…” Changes to “three similar compounds designated 1215 A1, A2 and A19, where the halogen was replaced by nitrile or trifluoromethyl groups, were synthesized.”

Author Response

Response to Reviewer 2 Comments

Reviewer 2

Inhibition of BR biosynthesis is an interesting field of research that has attracted a lot of effort during the last decades. In this context this review of small molecules exhibiting this activity will be useful for many readers of Molecules. This review is comprehensive and well organized, and the presented data is discussed properly. Thus, I recommend publishing this work after all points detailed below, and the attached file, have been addressed.

Response

Many thanks for the positive evaluation of our manuscript and for the helpful comments, which we have addressed as outlined below.

Changes in the manuscript were marked in yellow.

Comment 1

Line 62, 63, 62, 65 Each phrase in these lines requires references

Line 134, 137. Phrase needs reference

Response 1

References were included as suggested by reviewer 2.

Comment 2

Line 228. For recognition of BL by its receptor I think it is important to include the following reference. She, J., Z. Han, B. Zhou and J. Chai (2013). "Structural basis for differential recognition of brassinolide by its receptors." Protein & Cell 4(6): 475-482.

Response 2

We agree that this is an important article. We have cited it in the revised manuscript.

Comment 3

Line 275. This paragraph needs some references.

Response 3

As suggested, a reference for MYBL2 was added.

Comment 4

Line 370. I think that the following reference should be added to 153. Li, H., H. Wang and S. Jang (2017). "Rice Lamina Joint Inclination Assay." Bio-protocol 7: e2409.

Response 4

We agree that this is a helpful protocol, which we have cited in the revised manuscript.

Comment 5

Line 385. In this paragraph one reference is needed at least.

Response 5

In the revised manuscript we included yucaizol as an example. The corresponding references were of course also included.

Comment 6

Line 440. The phrase “Using such assays, the inhibitory effects of a number of steroidal and non-steroidal compounds on DET2 [169,170].” must be changed to “Using such assays, the inhibitory effects of a number of steroidal and non-steroidal compounds on DET2 [169,170] were determined”.

Response 6

The sentence was corrected as suggested.

Comment 7

Line 492. It reads Michaelis-Mention. Should be corrected, Same line “the” must deleted.

Response 7

“Michaelis-Mention” was corrected to “Michaelis-Menten”. As suggested, “the” was deleted.

Comment 8

Line 504. It should read “similar phenotype characterized by reduced hypocotyl…”

Response 8

The sentence was corrected as suggested.

Comment 9

Line 522. Results shown in Figure 5A and 5B are from literature or were measure in this work?? Specify in the text.

Line 535. Same as previous point.

Response 9

Figure 5 was prepared for this manuscript and the data have not been used elsewhere. We have now included the original data in Suppl. File 2 to show that the experiments were performed for this particular manuscript. Since these data have not been used before no reference can be given.

Comment 10

Line 564. Statements made in paragraph starting in this line require references

Line 566. “biosynthesis and responses and for manipulating BR levels…”. Delete

Response 10:

As suggested, the sentence was deleted and references were included in the paragraph.

Comment 11

Line 585. Figure 6. Triazol derivatives must be numbered in correlative form. Compounds 5 and 10 are the only ones with numbers. These numbers must be used in the text.

Response 11

The numbering corresponds to the original publication. We think that this improves traceability of the structures shown in our manuscript to the original publications. To make that clear we have now cited the original publication in the figure legend and mentioned that numbering refers to the original publications.

Comment 12

Line 649. The phrase “whose effected proteins act transcription factors in…..” should be rewritten

Response 12

The sentence was corrected.

Comment 13

Line 658. Figure 7. Compounds must be numbered properly.

Response 13

Numbering corresponds to the original publication. We have now mentioned that in the figure legend and cited the original publication. See also response to comment 11.

Comment 14

Line 662. “the reaction sequence for the example of Brz220.”  should be change to “the reaction sequence for the synthesis of Brz220.”

Response 14

The sentence was corrected as suggested.

Comment 15

Line 665. Number of compound 12 should adapted to right numeration.

Response 15

Numbering corresponds to the original publication. We have now mentioned that in the figure legend and cited the original publication. See also response to comment 11.

Comment 16

Line 689. It reads “Another series of BR biosynthesis inhibitors possessing a dioxolane ring….”. I think it is necessary to stress that this series are also triazol derivatives. For example, “Another series of BR biosynthesis inhibitors possessing a dioxolane ring, in addition to the triazol ring,…”

Response 16

We agree that it should be explicitly stated that the compounds of the YCZ-series are triazoles. We have adapted the sentence as suggested.

Comment 17

Line 694. Figure 8. Same as above, compounds should be numbered following the sequence as they appear in the text.

Response 17

Numbering corresponds to the original publication. We have now mentioned that in the figure legend and cited the original publication. See also response to comment 11.

Comment 18

Line 754. The phrase “This allowed using published data from synthesis of kentoconazole as a reference, which facilitated determination of the absolute configuration” should be rewritten

Response 18

We agree that the sentence was not clear. We have rewritten that part.

Comment 19

Line 962. Figure 11. Same as above, compounds should be numbered following the sequence as they appear in the text.

Response 19

Numbering corresponds to the original publication. We have now mentioned that in the figure legend and cited the original publication. See also response to comment 11.

Comment 20

Line 1001. “another means” should be changed “another way”

Response 20

The sentence was corrected as suggested.

Comment 21

Line 1017. Add the following reference “Liu, J., D. Zhang, X. Sun, T. Ding, B. Lei and C. Zhang (2017). "Structure-activity relationship of brassinosteroids and their agricultural practical usages." Steroids 124: 1-17.”

Response 21

We agree that this is an important publication. We have cited it the revised manuscript.

Comment 22

Line 1080. Numeration of compounds must be corrected.

Line 1086. Figure 15. Same as above, compounds should be numbered following the sequence as they appear in the text

Line 1102. Figure 16. Same as above, compounds should be numbered following the sequence as they appear in the text

Line 112. Figure 17. Same as above, compounds should be numbered following the sequence as they appear in the text

Response 22

Numbering corresponds to the original publication. We have now mentioned that in the figure legend and cited the original publication. See also response to comment 11.

Comment 23

Line 1118. Change to “Compound NSBR1 was synthesized….”

Response 23

The sentence was corrected as suggested.

Comment 24

Line 1215. It reads “three similar compounds designated 1215 A1, A2 and A19, where the halogen was replaced by nitrile or trifluoromethyl groups…” Changes to “three similar compounds designated 1215 A1, A2 and A19, where the halogen was replaced by nitrile or trifluoromethyl groups, were synthesized.”

Response 24

The sentence was corrected as suggested.